# Semi-Supervised Cross-Domain Imitation Learning

**Li-Min Chu**                                                    *andy27564@gmail.com*
*Department of Computer Science,*
*National Yang Ming Chiao Tung University,*
*Hsinchu, Taiwan*

**Kai-Siang Ma**                                                  *seanmama.cs10@nycu.edu.tw*
*Department of Computer Science,*
*National Yang Ming Chiao Tung University,*
*Hsinchu, Taiwan*

**Ming-Hong Chen**                                               *mhchen1224.cs12@nycu.edu.tw*
*Department of Computer Science,*
*National Yang Ming Chiao Tung University,*
*Hsinchu, Taiwan*

**Ping-Chun Hsieh**                                              *pinghsieh@nycu.edu.tw*
*Department of Computer Science,*
*National Yang Ming Chiao Tung University,*
*Hsinchu, Taiwan*

**Reviewed on OpenReview:** *https://openreview.net/forum?id=WARXnbJawZ*

## Abstract

Cross-domain imitation learning (CDIL) accelerates policy learning by transferring expert knowledge across domains, which is valuable in applications where the collection of expert data is costly. Existing methods are either *supervised*, relying on proxy tasks and explicit alignment, or *unsupervised*, aligning distributions without paired data, but often unstable. We introduce the Semi-Supervised CDIL (SS-CDIL) setting and propose the first algorithm for SS-CDIL with theoretical justification. Our method uses only offline data, including a small number of target expert demonstrations and some unlabeled imperfect trajectories. To handle domain discrepancy, we propose a novel cross-domain loss function for learning inter-domain state-action mappings and design an adaptive weight function to balance the source and target knowledge. Experiments on MuJoCo and Robosuite show consistent gains over the baselines, demonstrating that our approach achieves stable and data-efficient policy learning with minimal supervision. Our code is available at `https://github.com/NYCU-RL-Bandits-Lab/CDIL`.

## 1 Introduction

In contrast to reinforcement learning (RL), which requires extensive environment interactions and carefully designed reward functions, imitation learning (IL) offers a compelling alternative by enabling efficient behavior acquisition through direct imitation of expert demonstrations, thus avoiding the need for complex reward engineering or exhaustive exploration. IL has demonstrated remarkable success in areas such as robot control (Mandlekar et al., 2022), autonomous driving (Le Mero et al., 2022), and visual navigation (Nguyen et al., 2019), where access to expert data facilitates the development of high-performance control policies. Despite these advantages, conventional IL methods commonly rely on the assumption that the expert data and the target environment share the same domain characteristics, a condition that is rarely satisfied in practice. In real-world scenarios, collecting expert demonstrations in target domains such as physical robotics

or self-driving cars is often prohibitively expensive or hazardous. In contrast, simulated environments, although capable of providing large-scale data, typically differ in their underlying dynamics and state-action representations.

Cross-domain imitation learning (CDIL) addresses this challenge by transferring expert knowledge from a source domain to a target domain with distinct dynamics or state-action spaces, thereby improving efficiency and reducing reliance on target-domain data. Existing CDIL approaches can be broadly categorized based on the level of supervision they require: (1) *Supervised* methods typically rely on paired or unpaired demonstrations in various proxy tasks to explicitly establish correspondences between domains. For example, (Sermanet et al., 2018; Liu et al., 2020) leverage paired visual trajectories for time-contrastive or state-translation learning, while (Raychaudhuri et al., 2021) align demonstrations via state-distribution matching. (2) In contrast, *unsupervised* methods avoid paired data by aligning distributions or learning invariant representations, such as adversarial domain adaptation (Kim et al., 2020; Zolna et al., 2021) and optimal transport-based alignment (Fickinger et al., 2022). These methods often assume an isomorphism between the source and target domains; however, such an isomorphism can generally be satisfied in multiple ways (possibly infinitely many), resulting in ambiguity in alignment and potential instability. In general, supervised approaches are constrained by the cost of collecting paired trajectories, unsupervised approaches often suffer from unstable transfer, and domain-specific solutions lack generality.

In this paper, we propose to study *Semi-Supervised CDIL*, which achieves cross-domain transfer by leveraging source-domain (imperfect) demonstrations and requires only minimal target-domain supervision by combining a small number of target-domain expert demonstrations along with imperfect target-domain trajectories, without the need for additional proxy tasks. This semi-supervised CDIL setting offers a promising balance between supervision cost and transfer capability. However, this setting remains largely unexplored, motivating the need for a principled framework.

To address this, we propose AdaptDICE, the first algorithmic framework for Semi-Supervised CDIL that extends the single-domain distribution correction estimation (DICE) to achieve knowledge transfer across domains with discrepancies in transition dynamics, state space, and action space. AdaptDICE leverages source-domain knowledge together with limited offline target-domain data, enabling stable policy learning without requiring paired trajectories or assumptions on isomorphism. The core of our approach consists of three components:

- **Cross-domain mapping loss**: AdaptDICE introduces a novel mapping loss that learns cross-domain mapping functions by minimizing the Bellman error within the mapped source-domain space, thereby bridging domain gaps while preserving source-domain knowledge. The learned cross-domain mappings enable knowledge transfer of source-domain density ratios to the target domain.

- **Hybrid density ratio for cross-domain policy extraction:** AdaptDICE derives the target-domain policy by combining density ratios of both domains via the cross-domain mappings. The resulting hybrid density ratio serves as the mechanism that facilitates transfer across domains in imitation learning.

- **Adaptive weighting**: The hybrid density ratio is further modulated by an adaptive weighting factor, $\beta(t)$, defined as a smooth function of the relative estimation errors between the source and target density ratios. This adaptive mechanism dynamically balances the contributions of both domains, enabling stable adaptation without manual hyperparameter tuning.

Through theoretical analysis, we establish the convergence guarantee of AdaptDICE for the underlying density-ratio estimation, achieving a polynomially decaying error bound, with the expected error adaptively bounded by the better of the two domains. Through extensive experiments, we show that AdaptDICE achieves consistently better performance than various baseline methods in challenging offline cross-domain scenarios (with as few as only 1 labeled target-domain expert trajectory) across multiple standard benchmark tasks in MuJoCo and RoboSuite. We also provide an ablation study to demonstrate the benefits of hybrid density ratios for effective cross-domain transfer. These results highlight the effectiveness and wide applicability of AdaptDICE under minimal supervision.

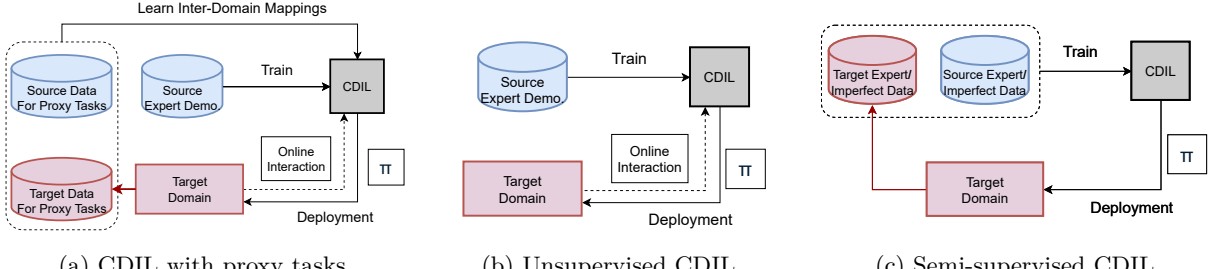

(a) CDIL with proxy tasks.  (b) Unsupervised CDIL.  (c) Semi-supervised CDIL.

Figure 1: An illustration of CDIL formulations: (a) CDIL with proxy tasks utilizes annotated target demonstrations through paired or unpaired proxy tasks, trading off accuracy and data efficiency. (b) Unsupervised CDIL relies only on source experts and unlabeled target data and typically requires assumptions on domain similarity (*e.g.,* isomorphism) and can suffer from ineffective transfer. (c) Semi-supervised CDIL combines limited labeled target data with unlabeled trajectories, balancing supervision cost and transferability.

## 2 Preliminaries

### 2.1 Cross-Domain Imitation Learning

In this section, we introduce the fundamental concepts and notation for the general CDIL, building on the standard Markov Decision Process (MDP) framework. For a set $\mathcal{X}$, let $\Delta(\mathcal{X})$ denote the set of all probability distributions over $\mathcal{X}$. We model each environment as an infinite-horizon discounted MDP, defined as $\mathcal{M} := (\mathcal{S}, \mathcal{A}, P, \mu, \gamma)$, where $\mathcal{S}$ is the state space, $\mathcal{A}$ is the action space, $P : \mathcal{S} \times \mathcal{A} \to \Delta(\mathcal{S})$ represents the transition function, giving the probability $P(s_{t+1}|s_t, a_t)$ of a transition from state $s_t$ to $s_{t+1}$ by taking action $a_t$ at time $t$, $\mu \in \Delta(\mathcal{S})$ is the initial state distribution, and $\gamma \in [0, 1)$ is the discount factor. A policy $\pi : \mathcal{S} \to \Delta(\mathcal{A})$ maps states to distributions over actions. For a given policy $\pi$, the occupancy measure is defined as $d^\pi(s, a) := (1 - \gamma)\mathbb{E}_{s_0 \sim \mu, P, \pi}[\sum_{t=0}^{\infty} \gamma^t \mathbb{P}(s_t = s, a_t = a|s_0)]$.[1]

In the canonical CDIL setting, the learning involves two domains, referred to as *source domain* (src) and *target domain* (tar), defined respectively as $\mathcal{M}_{\text{src}} := (\mathcal{S}_{\text{src}}, \mathcal{A}_{\text{src}}, P_{\text{src}}, \mu_{\text{src}}, \gamma)$ and $\mathcal{M}_{\text{tar}} := (\mathcal{S}_{\text{tar}}, \mathcal{A}_{\text{tar}}, P_{\text{tar}}, \mu_{\text{tar}}, \gamma)$. Notably, the source and target domains can differ significantly in transition dynamics, state space, and action space. We assume that both MDPs share the same discount factor $\gamma$. In CDIL, the information available to the imitator can be summarized as follows:

- **Target domain**: The imitator operates in $\mathcal{M}_{\text{tar}}$ and aims to mimic a target-domain expert policy $\pi_{\text{E}} : \mathcal{S}_{\text{tar}} \to \Delta(\mathcal{A}_{\text{tar}})$ by learning from either online interactions in the target domain or a pre-collected target-domain dataset that can consist of either expert and non-expert demonstrations. In CDIL, both the online interaction budget and the size of the target-domain dataset are presumed to be small.

- **Source domain**: In CDIL, due to the limited target-domain data available, the imitator also leverages auxiliary information from the source domain (*e.g.,* via a pre-collected source-domain dataset $\mathcal{D}_{\text{src}}$).

The goal of the imitator is to learn a policy $\pi^*_{\text{tar}} : \mathcal{S}_{\text{tar}} \to \Delta(\mathcal{A}_{\text{tar}})$ that can effectively mimic the expert policy based on the data from both domains. In the CDIL literature, there exist two commonly adopted problem settings:

- **CDIL with proxy tasks**: This supervised setting assumes the existence of *proxy* or *alignment tasks* with expert demonstrations in both source and target domains (Kim et al., 2020; Raychaudhuri et al., 2021) as a form of supervision. For instance, in robot control, primitive tasks like walking or jumping can serve as proxies for more complex target tasks. Cross-domain knowledge transfer is achieved via domain alignment, *i.e.,* by learning cross-domain state-action mappings from these auxiliary expert data, either paired or unpaired across domains (Figure 1(a)). This supervised setting enables more accurate alignment and stable transfer but requires substantial data collection or annotation.

---

[1]In the RL literature, the occupancy measure $d^\pi$ is also called the discounted state-action visitation distribution.

- **Unsupervised CDIL**: This setting relies primarily on source-domain expert data and uses no target-domain demonstrations, hence considered "unsupervised" (Fickinger et al., 2022; Franzmeyer et al., 2022). Learning cross-domain mappings still requires online interaction with the target environment (Figure 1(b)). While avoiding reliance on target-domain experts, unsupervised CDIL algorithms often assume strong domain isomorphism and can suffer from instability or misalignment under substantial domain discrepancies.

## 2.2 Regularized Distribution Matching

In an offline setup, (single-domain) imitation learning can be formulated as a *regularized distribution matching* problem, which minimizes the discrepancy in occupancy measure between the expert and the imitator under behavior regularization to overcome extrapolation error in offline learning. Specifically, one notable offline IL formulation is introduced by DemoDICE (Kim et al., 2022) and substantiated by the following objective function:

$$\pi^* = \text{argmax}_\pi -D_{\text{KL}}(d^\pi \parallel d^{\text{E}}) - \alpha D_{\text{KL}}(d^\pi \parallel d^{\text{U}}), \tag{1}$$

where $\alpha > 0$ is the regularization coefficient and $d^{\text{E}}$ and $d^{\text{U}}$ denote the underlying occupancy measures of the expert and imperfect datasets, respectively. As the objective function in (1) is generally non-convex in $\pi$, one can use $d^\pi$ instead of $\pi$ as the decision variables and thereby reformulate (1) as a *convex constrained optimization* problem that minimizes the KL divergence between the imitator's and expert's occupancy measures under a set of Bellman flow conservation constraints (*i.e.,* one equality constraint for each state). By resorting to the dual problem and strong duality and introducing a numerically-stable transformation, we can arrive at an equivalent unconstrained problem:

$$\min_\nu \ L(\nu; r) := (1 - \gamma) \mathbb{E}_{s \sim \mu}[\nu(s)] + (1 + \alpha) \log \mathbb{E}_{(s,a) \sim d^{\text{U}}} \left[ \exp \left( \frac{A_\nu(s, a)}{1 + \alpha} \right) \right], \tag{2}$$

where $\{\nu(s)\}_{s \in \mathcal{S}}$ are the Lagrange multipliers (one for each constraint), $r(s, a) := \log(d^{\text{E}}(s, a)/d^{\text{U}}(s, a))$ is the logarithmic density ratio, and $A_\nu(s, a) := r(s, a) + \gamma \mathbb{E}_{s' \sim P(\cdot|s,a)}[\nu(s')] - \nu(s)$. We let $\nu^*$ denote the optimizer of (2). Notably, one can interpret $r(s, a)$ as the *pseudo reward* and $\nu(s)$ as the *pseudo value function* in offline IL, and hence $A_\nu(s, a)$ can be viewed as the *advantage function* under $r$ and $\nu$.

Once the dual problem in (2) is solved and the optimal dual variables $\nu^*$ are obtained, we can extract the optimal primal variables $d^{\pi^*}$ and the corresponding optimal policy $\pi^*$ as follows: (i) We first define a density ratio $w^*(s, a) := d^*(s, a)/d^{\text{U}}(s, a)$. The duality gives us that for each $(s, a)$,

$$w^*(s, a) = \exp \left( \frac{A_{\nu^*}(s, a)}{1 + \alpha} \right). \tag{3}$$

(ii) Based on the derived $w^*(s, a)$ in (3), one can extract $\pi^*$ through weighted behavior cloning, which minimizes $\mathbb{E}_{(s,a) \sim d^{\text{U}}} \left[ w^*(s, a) \log \pi(a|s) \right]$ over $\pi$.

## 3 Methodology

In this section, we formally describe the proposed formulation of Semi-Supervised CDIL. Then, we present *AdaptDICE* (Adaptive Cross-Domain Imitation Learning with DIstribution Correction Estimation), the first algorithmic framework for Semi-Supervised CDIL. We start by introducing a prototypic algorithm based on AdaptDICE designed for efficient knowledge transfer across domains and analyze the convergence of the proposed algorithm in the tabular case. Subsequently, we propose an adaptive function that dynamically balances source- and target-domain knowledge to accommodate diverse domain disparities and then provide a practical implementation beyond the tabular case.

## 3.1 Semi-Supervised CDIL

As described in Section 2.1, fully supervised CDIL enables accurate cross-domain alignment but requires substantial target-domain demonstrations, while unsupervised CDIL avoids this cost but often fails under

---

**Algorithm 1** AdaptDICE

---

**Require:** Source-domain $Q_{\mathrm{src}}$ and $w^{\mathrm{src}}$ pretrained with DemoDICE, weight function $\beta : \mathbb{N} \to [0, 1]$, step size $\eta$, target-domain offline dataset $\mathcal{D}_{\mathrm{tar}}^{\mathrm{U}} = \{(s, a, s')\}$, expert dataset $\mathcal{D}_{\mathrm{tar}}^{\mathrm{E}} = \{(s, a)\} \subseteq \mathcal{D}_{\mathrm{tar}}^{\mathrm{U}}$
 1: Initialize discriminator network $c_{\mathrm{tar}} : \mathcal{S}_{\mathrm{tar}} \times \mathcal{A}_{\mathrm{tar}} \to [0, 1]$
 2: Update the discriminator by (10):   $c_{\mathrm{tar}} \leftarrow \arg\min_{c_{\mathrm{tar}}} J_{c_{\mathrm{tar}}}(\mathcal{D}_{\mathrm{tar}}^{\mathrm{E}}, \mathcal{D}_{\mathrm{tar}}^{\mathrm{U}})$
 3: Compute $r_{\mathrm{tar}}(s, a) = -\log(1/c_{\mathrm{tar}}(s, a) - 1)$ for $(s, a) \in \mathcal{D}_{\mathrm{tar}}^{\mathrm{U}}$
 4: Initialize $G^{(0)}, H^{(0)}, \nu_{\mathrm{tar}}^{(0)}, \pi^{(0)}$
 5: **for** iteration $t = 1, \ldots, T$ **do**
 6:     Sample $\mathcal{D}_{\mathrm{tar}}$ of size $N_{\mathrm{tar}}$ from the offline dataset $\mathcal{D}_{\mathrm{tar}}^{\mathrm{U}}$
 7:     Compute $\beta(t)$ by (18)
 8:     Update the mapping functions by (4): $G^{(t)}, H^{(t)} \leftarrow \arg\min_{G,H} L_{\mathrm{MAP}}(G, H; r_{\mathrm{tar}}, Q_{\mathrm{src}}, \pi^{(t-1)}, \mathcal{D}_{\mathrm{tar}})$
 9:     Update the pseudo value by (5): $\nu_{\mathrm{tar}}^{(t)} \leftarrow \nu_{\mathrm{tar}}^{(t-1)} - \eta \nabla_{\nu_{\mathrm{tar}}} L_{\mathrm{DICE}}(\nu_{\mathrm{tar}}^{(t-1)}; r_{\mathrm{tar}}, \pi^{(t-1)}, \mathcal{D}_{\mathrm{tar}})$
 10:     Update the policy by (9):   $\pi^{(t)} \leftarrow \arg\min_{\pi} L_{\mathrm{BC}}(\pi; G^{(t)}, H^{(t)}, \beta(t), w_{\mathrm{src}}, w_{\mathrm{tar}}^{(t)}, \mathcal{D}_{\mathrm{tar}})$
 11: **return** Target-domain policy $\pi_{\mathrm{tar}}^{(T)}$

---

large domain discrepancies. In practice, target domains usually provide few expert demonstrations, whereas imperfect or suboptimal data are easier to obtain.

To address this, we formalize Semi-Supervised CDIL, assuming access to a small labeled subset of expert demonstrations and the rest as unlabeled trajectories. Labeled data provides minimal supervision, while unlabeled data captures target dynamics (Figure 1(c)), reducing annotation cost and avoiding instability. The information available to the imitator in Semi-Supervised CDIL can be summarized as follows:

- **Target domain**: In this paper, we focus on the offline setting, where the imitator has access to a pre-collected expert dataset $\mathcal{D}_{\mathrm{tar}}^{\mathrm{E}}$ of $(s, a, s')$ tuples, which are generated by $\pi_{\mathrm{tar}}^{\mathrm{E}}$ and follow the occupancy measure $d_{\mathrm{tar}}^{\mathrm{E}}$, as well as an imperfect dataset $\mathcal{D}_{\mathrm{tar}}^{\mathrm{I}}$, which is generated by policies with unknown degrees of optimality. We define the union dataset as $\mathcal{D}_{\mathrm{tar}}^{\mathrm{U}} := \mathcal{D}_{\mathrm{tar}}^{\mathrm{E}} \cup \mathcal{D}_{\mathrm{tar}}^{\mathrm{I}}$ and let $d^{\mathrm{U}}$ denote the corresponding mixture of occupancy measures. In CDIL, $\mathcal{D}_{\mathrm{tar}}^{\mathrm{E}}$ is presumed to be scarce (*e.g.,* as few as only one trajectory), and hence cross-domain transfer is especially needed.

- **Source domain**: Due to the limited target-domain data, the imitator leverages an auxiliary source-domain dataset $\mathcal{D}_{\mathrm{src}}$, which can consist of both expert and non-expert demonstrations in the source domain. With $\mathcal{D}_{\mathrm{src}}$, one can also leverage an offline IL algorithm to pre-train policies or other relevant networks for cross-domain transfer. In particular, a source-domain pseudo Q-function $Q_{\mathrm{src}}$ can be obtained as a pre-trained critic, which serves as a transferable component and will be utilized later in our framework to guide cross-domain adaptation.

### 3.2   Proposed Algorithm

In this subsection, we formally present AdaptDICE and describe its key components. The pseudo code is provided in Algorithm 1. The algorithm leverages a target-domain offline dataset $\mathcal{D}_{\mathrm{tar}}^{\mathrm{U}} = \{(s, a, s')\}$ alongside pre-trained source-domain models, to learn an optimal policy for the target domain.

Before describing the key components of AdaptDICE, we first provide an overview of its learnable components. The algorithm begins by training a discriminator $c_{\mathrm{tar}}$ (Lines 1-2) to induce a pseudo reward, representing a logarithmic density ratio $r_{\mathrm{tar}}$ (Line 3), which guides subsequent optimization. AdaptDICE then iteratively updates three key modules: (1) a pseudo value function $\nu_{\mathrm{tar}}^{(t)}$ updated via DICE-based gradient descent; (2) mapping networks $G^{(t)}$ and $H^{(t)}$ that align source and target domains by minimizing a Bellman error; and (3) the target policy $\pi_{\mathrm{tar}}^{(t)}$ refined through weighted behavior cloning with adaptive weight $\beta(t)$. These updates occur within the loop (Lines 5-11), facilitating robust cross-domain knowledge transfer. Notably, this transfer is achieved by querying source signals ($Q_{\mathrm{src}}$ and $w_{\mathrm{src}}$) on the mapped tuples $(G(s), H(s, a))$, effectively utilizing them as informative priors. Meanwhile, the target density ratio $w_{\mathrm{tar}}$ and the mappings $(G, H)$ are learned directly from the target-domain data to avoid degenerate mappings.

We now describe the three components updated inside the training loop (Lines 8-10) in the following order: the cross-domain mapping loss, the DICE loss, and the cross-domain policy extraction. The pseudo-reward computation, which is performed once before the loop, is detailed at the end.

**Cross-Domain Mapping Loss.** The mapping loss $L_{\mathrm{MAP}}$ (Line 8) learns cross-domain mapping functions $G : \mathcal{S}_{\mathrm{tar}} \to \mathcal{S}_{\mathrm{src}}$ and $H : \mathcal{S}_{\mathrm{tar}} \times \mathcal{A}_{\mathrm{tar}} \to \mathcal{A}_{\mathrm{src}}$, which align source- and target-domain transitions such that target-domain state-action pairs can be evaluated under the source-domain critic. We learn $G$ and $H$ by minimizing the following loss, which enforces Bellman consistency in the mapped source-domain space:

$$
\begin{aligned}
L_{\mathrm{MAP}}(G, H; r_{\mathrm{tar}}, &Q_{\mathrm{src}}, \pi, \mathcal{D}_{\mathrm{tar}}) := \\
&\mathbb{E}_{(s,a,s')\sim\mathcal{D}_{\mathrm{tar}}} \big| r_{\mathrm{tar}}(s,a) + \gamma \mathbb{E}_{a'\sim\pi(s')} \left[ Q_{\mathrm{src}}(G(s'), H(s',a')) \right] - Q_{\mathrm{src}}(G(s), H(s,a)) \big|,
\end{aligned}
\tag{4}
$$

where $r_{\mathrm{tar}}$ is the target-domain pseudo reward (computed once before the training loop; detailed at the end of this subsection), and $Q_{\mathrm{src}} := r_{\mathrm{src}}(s,a) + \gamma \mathbb{E}_{s'\sim P_{\mathrm{src}}(\cdot|s,a)}[\nu_{\mathrm{src}}(s')]$ is source-domain pre-trained optimal pseudo Q-function. By minimizing this loss, we obtain mappings $G^*$ and $H^*$ that effectively bridge the domain gap, ensuring that the source-domain knowledge can be reliably adapted to the target domain without direct modification to $Q_{\mathrm{src}}$. In the tabular setting, $L_{\mathrm{MAP}}$ yields optimal mappings by solving the finite state-action space, minimizing Bellman error over $\mathcal{D}_{\mathrm{tar}}$.

**DICE Loss.** In AdaptDICE, the loss $L_{\mathrm{DICE}}$ (Line 9) is employed to learn the pseudo value function for the density ratio $w^*$ based on the target-domain data. Specifically, it updates $\nu_{\mathrm{tar}}$ through gradient-based optimization on the following loss on offline samples $\mathcal{D}_{\mathrm{tar}}$, *i.e.*,

$$
L_{\mathrm{DICE}}(\nu_{\mathrm{tar}}; r_{\mathrm{tar}}, \pi, \mathcal{D}_{\mathrm{tar}}) := (1-\gamma)\mathbb{E}_{s\sim\mu_{\mathrm{tar}}}[\nu_{\mathrm{tar}}(s)] + (1+\alpha) \cdot \log \mathbb{E}_{(s,a)\sim\mathcal{D}_{\mathrm{tar}}} \left[ \exp\left( \frac{A_{\nu_{\mathrm{tar}}}(s,a)}{1+\alpha} \right) \right],
\tag{5}
$$

where $\mu_{\mathrm{tar}}$ is the initial state distribution induced from the dataset $\mathcal{D}_{\mathrm{tar}}^{\mathrm{U}}$.

**Cross-Domain Policy Extraction.** By integrating the mapping function mentioned before, we define the cross-domain density ratio $w_{\mathrm{cross}}$ that is combined with the pre-trained source-domain density ratio $w_{\mathrm{src}}$ and the learned target-domain density ratio $w_{\mathrm{tar}}$:

$$
w_{\mathrm{cross}}(s,a) := \beta\, w_{\mathrm{src}}(G(s), H(s,a)) + (1-\beta)\, w_{\mathrm{tar}}(s,a),
\tag{6}
$$

where $\beta \in [0,1]$ is a parameter that balances the contribution of the source and target domains. The density ratios for the source and target domains are defined as

$$
w_{\mathrm{src}}(G(s), H(s,a)) := \exp\left( \frac{A_{\nu_{\mathrm{src}}}(G(s), H(s,a))}{1+\alpha} \right),
\tag{7}
$$

$$
w_{\mathrm{tar}}(s,a) := \exp\left( \frac{A_{\nu_{\mathrm{tar}}}(s,a)}{1+\alpha} \right),
\tag{8}
$$

where $\nu_{\mathrm{src}}$ denotes the pre-trained source-domain pseudo value function and $\nu_{\mathrm{tar}}$ is the learned target-domain pseudo value function. This cross-domain density ratio is used in the behavioral cloning loss $L_{\mathrm{BC}}$ (Line 10):

$$
L_{\mathrm{BC}}(\pi; G, H, \beta, w_{\mathrm{src}}, w_{\mathrm{tar}}, \mathcal{D}_{\mathrm{tar}}) := -\mathbb{E}_{(s,a)\sim\mathcal{D}_{\mathrm{tar}}} \left[ w_{\mathrm{cross}}(s,a) \cdot \log \pi(a|s) \right].
\tag{9}
$$

is minimized to learn an optimal policy in a tabular setting with a finite state-action space. In Algorithm 1, we employ an adaptive weighting factor $\beta(t)$ (Line 7) whose exact form is detailed in Section 3.4, enabling the weighted behavioral cloning loss during training without manual tuning of $\beta(t)$.

As mentioned earlier, before entering the training loop, we first train a discriminator to obtain the target-domain pseudo-reward as follows:

**Pseudo-Reward Computation.** A discriminator network $c_{\text{tar}} : \mathcal{S}_{\text{tar}} \times \mathcal{A}_{\text{tar}} \to [0,1]$ is trained to distinguish expert state-action pairs from $\mathcal{D}_{\text{tar}}^{\text{E}}$ against the full target-domain dataset $\mathcal{D}_{\text{tar}}^{\text{U}}$, minimizing the binary cross-entropy loss $J_{c_{\text{tar}}}$ (Line 2):

$$J_{c_{\text{tar}}}(\mathcal{D}_{\text{tar}}^{\text{E}}, \mathcal{D}_{\text{tar}}^{\text{U}}) := - \left( \mathbb{E}_{(s,a)\sim\mathcal{D}_{\text{tar}}^{\text{E}}}[\log c_{\text{tar}}(s,a)] + \mathbb{E}_{(s,a)\sim\mathcal{D}_{\text{tar}}^{\text{U}}}[\log(1 - c_{\text{tar}}(s,a))] \right). \tag{10}$$

The target-domain pseudo-reward (Line 3) is constructed as

$$r_{\text{tar}}(s,a) := -\log(1/c_{\text{tar}}(s,a) - 1), \tag{11}$$

which approximates the true logarithmic density ratio $\log(d_{\text{tar}}^{\text{E}}(s,a)/d_{\text{tar}}^{\text{U}}(s,a))$.

## 3.3 Convergence Analysis

In this subsection, we theoretically establish the convergence of AdaptDICE, focusing on how the cross-domain density ratio $w_{\text{cross}}^{(t)}(s,a)$ approaches the optimal density ratio $w_{\text{tar}}^*(s,a)$. To measure this convergence, we define the density ratio error:

$$|w_{\text{cross}}^{(t)}(s,a) - w_{\text{tar}}^*(s,a)|. \tag{12}$$

We then derive an upper bound on this error, highlighting the role of the weighting factor $\beta(t)$ in balancing source and target domain contributions to achieve efficient convergence.

Building on the formulation in Section 3.2, the cross-domain density ratio in AdaptDICE is given by:

$$w_{\text{cross}}^{(t)}(s,a) := \beta(t)w_{\text{src}}(G^{(t)}(s), H^{(t)}(s,a)) + (1-\beta(t))w_{\text{tar}}^{(t)}(s,a), \tag{13}$$

where $w_{\text{src}}(G^{(t)}(s), H^{(t)}(s,a)) := \exp\left(A_{\nu_{\text{src}}}(G^{(t)}(s), H^{(t)}(s,a))/(1+\alpha)\right)$ is the pre-trained source-domain density ratio, $w_{\text{tar}}^{(t)}(s,a) := \exp(A_{\nu_{\text{tar}}^{(t)}}(s,a)/(1+\alpha))$ is the target-domain density ratio at iteration $t$, and $\beta(t) : \mathbb{N} \to [0,1]$ is a weighting factor balancing the contributions of the source and target domains. The mapping functions $G$ and $H$ denote the learned state and action mappings that align target-domain samples with the source-domain representation. To quantify the convergence of $w_{\text{cross}}^{(t)}(s,a)$ to the optimal density ratio $w_{\text{tar}}^*(s,a)$, we define the pointwise density ratio errors:

$$\Delta w_{\text{src}}^{(t)}(s,a) := |w_{\text{src}}(G^{(t)}(s), H^{(t)}(s,a)) - w_{\text{tar}}^*(s,a)|, \quad \Delta w_{\text{tar}}^{(t)}(s,a) := |w_{\text{tar}}^{(t)}(s,a) - w_{\text{tar}}^*(s,a)|, \tag{14}$$

and their expected errors over the target-domain distribution $\mathcal{D}_{\text{tar}}$:

$$\Delta \bar{w}_{\text{src}}^{(t)} := \mathbb{E}_{(s,a)\sim\mathcal{D}_{\text{tar}}}|w_{\text{src}}(G^{(t)}(s), H^{(t)}(s,a)) - w_{\text{tar}}^*(s,a)|, \ \Delta \bar{w}_{\text{tar}}^{(t)} := \mathbb{E}_{(s,a)\sim\mathcal{D}_{\text{tar}}}|w_{\text{tar}}^{(t)}(s,a) - w_{\text{tar}}^*(s,a)|. \tag{15}$$

For convergence analysis, we define the optimal solution set of $\nu_{\text{tar}}$ as $S_{\text{tar}}^* := \{\nu_{\text{tar}}^* + C \cdot \mathbf{1}_{\mathcal{S}_{\text{tar}}} \mid C \in \mathbb{R}\}$ where $\mathbf{1}_{\mathcal{S}_{\text{tar}}}$ is the all-ones vector over $\mathcal{S}_{\text{tar}}$, and the orthogonal projection $\Pi_{S_{\text{tar}}^*}(\nu_{\text{tar}}) := \arg\min_{y \in S_{\text{tar}}^*} \|\nu_{\text{tar}} - y\|_2$.

We next provide the convergence guarantee of AdaptDICE, whose proof is deferred to Appendix C.

**Theorem 1.** [Upper Bound of Cross-Domain Density Ratio Error Under AdaptDICE] *Under AdaptDICE, with learning rate $\eta \leq 1/L_f$, for each $(s,a)$, the cross-domain density ratio error is bounded as follows:*

$$|w_{cross}^{(t)}(s,a) - w_{tar}^*(s,a)| \leq \beta(t)\Delta w_{src}^{(t)}(s,a) + (1-\beta(t))\left[C_w w_{tar}^*(s,a) \exp\left(\frac{C_w}{\sqrt{t}}\|\nu_{tar}^{(0)} - \nu_{tar}^*\|_2\right)\frac{\|\nu_{tar}^{(0)} - \nu_{tar}^*\|_2}{\sqrt{t}}\right], \tag{16}$$

*where $L_f$ is the smoothness constant of $L_{DICE}$, $\nu_{tar}^* := \Pi_{S_{tar}^*}(\nu_{tar}^{(0)})$, and $C_w$ is a constant. Moreover, by selecting $\beta(t)$ as*

$$\beta(t) = \begin{cases} 0, & \text{if } \Delta\bar{w}_{tar}^{(t)} \leq \Delta\bar{w}_{src}^{(t)} \\ 1, & \text{otherwise} \end{cases}$$

*the expected cross-domain density ratio error satisfies*

$$\mathbb{E}_{(s,a)\sim\mathcal{D}_{tar}}|w_{cross}^{(t)}(s,a) - w_{tar}^*(s,a)| \leq \min(\Delta\bar{w}_{src}^{(t)}, \Delta\bar{w}_{tar}^{(t)}). \tag{17}$$

**Remark.** The bound in Theorem 1 demonstrates the effectiveness of AdaptDICE in adaptively combining source- and target-domain density ratios. By choosing $\beta(t)$ to favor the domain with the smaller expected error, AdaptDICE achieves an error bound no worse than the better of $\Delta \bar{w}_{\text{src}}^{(t)}$ or $\Delta \bar{w}_{\text{tar}}^{(t)}$. When the source-domain density ratio is more accurate (*i.e.,* $\Delta \bar{w}_{\text{src}}^{(t)} < \Delta \bar{w}_{\text{tar}}^{(t)}$), setting $\beta(t) = 1$ leverages the pre-trained source model to accelerate convergence compared to relying solely on the target-domain density ratio. This adaptability is particularly valuable in scenarios with limited target-domain data, as it enables efficient knowledge transfer while mitigating errors due to domain discrepancies.

### 3.4 Design Choice of Weighting Factor $\beta(t)$

Recall that Theorem 1 establishes a discrete selection rule for the weighting factor $\beta(t)$ that yields an optimal expected error bound. However, such a hard-switching mechanism may cause undesirable oscillations in practice when the source and target errors are comparable. To achieve smoother and more stable adaptation behavior, AdaptDICE employs a continuous weighting scheme that interpolates between the source and target density ratios. Specifically, we define $\beta(t)$ as a function of the expected density ratio errors:

$$\beta(t) = \frac{\frac{1}{\Delta \bar{w}_{\text{src}}^{(t)}}}{\frac{1}{\Delta \bar{w}_{\text{src}}^{(t)}} + \frac{1}{\Delta \bar{w}_{\text{tar}}^{(t)}}}, \tag{18}$$

where $\Delta \bar{w}_{\text{src}}^{(t)}, \Delta \bar{w}_{\text{src}}^{(t)}$ denote the expected estimation errors of the mapped source-domain and target-domain density ratios (Equation (15)). This formulation adaptively adjusts $\beta(t)$ according to the relative reliability of the two estimators: when the source ratio is more accurate ($\Delta \bar{w}_{\text{src}}^{(t)} < \Delta \bar{w}_{\text{tar}}^{(t)}$), $\beta(t) > 0.5$ and smoothly increases toward 1 as the source estimator becomes more reliable, emphasizing source-domain information; conversely, when the target estimator is more accurate, $\beta(t)$ decreases toward 0, shifting the emphasis to the target domain. Such a smooth transition avoids abrupt switches between domains and maintains stable weighting during training.

While this adaptive weighting does not alter the theoretical convergence rate established in Theorem 1, It offers notable practical advantages. First, continuous interpolation mitigates oscillations in $\beta(t)$ when the source and target estimators exhibit similar errors, leading to smoother updates of $w_{\text{cross}}^{(t)}$ and more stable policy optimization. Moreover, it enables a gradual transition of reliance from the source to the target domain, preventing abrupt degradation and improving robustness when the target-domain data are limited or noisy. These properties collectively improve the empirical stability and reliability of AdaptDICE in diverse cross-domain adaptation scenarios.

### 3.5 Practical Implementation

In this subsection, we present the practical aspects of AdaptDICE, including optimization of the mapping functions, the implementation of the adaptive function, and the gradient-based training of loss functions.

**Cross-Domain Mapping Optimization.** To align the source and target domains with potentially distinct state and action spaces, we optimize mapping functions $G : \mathcal{S}_{\text{tar}} \to \mathcal{S}_{\text{src}}$ and $H : \mathcal{S}_{\text{tar}} \times \mathcal{A}_{\text{tar}} \to \mathcal{A}_{\text{src}}$ using a cross-domain loss $L_{\text{MAP}}$ in Equation (4) that ensures Bellman consistency across the source and target domains. To ensure that $G$ and $H$ map target-domain states and actions to a source-domain feasible region, we adopt normalizing flows, following (Brahmanage et al., 2023) in the action-constrained RL literature (Hung et al., 2025), to map the outputs of $G$ and $H$ into the source-domain feasible region. Specifically, target-domain states and actions are first mapped by fully-connected layers into a latent dummy region (*e.g.,* a hypercube $[0, 1]^N$), then transformed by a normalizing flow into the feasible source-domain region. The flow model is pre-trained entirely offline using source-domain feasible samples and kept fixed during target-domain learning, with only the preceding network layers updated.

**Adaptive Function Implementation.** The adaptive weighting function $\hat{\beta}(t) : \mathbb{N} \to [0, 1]$ prioritizes the domain whose density ratio is closer to the optimal target-domain ratio $w^*_{\text{tar}}(s, a)$. It is computed as:

$$\hat{\beta}(t) := \frac{\frac{1}{\Delta\hat{w}^{(t)}_{\text{src}}}}{\frac{1}{\Delta\hat{w}^{(t)}_{\text{src}}} + \frac{1}{\Delta\hat{w}^{(t)}_{\text{MA}}}}, \tag{19}$$

where the empirical error estimates are:

$$\Delta\hat{w}^{(t)}_{\text{src}} := \mathbb{E}_{(s,a)\sim\mathcal{D}_{\text{tar}}} \left| w_{\text{src}}(G^{(t)}(s), H^{(t)}(s, a)) - \hat{w}^*_{\text{tar}}(s, a) \right|, \tag{20}$$

$$\Delta\hat{w}^{(t)}_{\text{tar}} := \mathbb{E}_{(s,a)\sim\mathcal{D}_{\text{tar}}} \left| w^{(t-1)}_{\text{tar}}(s, a) - \hat{w}^*_{\text{tar}}(s, a) \right|. \tag{21}$$

Since the true $w^*_{\text{tar}}(s, a)$ is unavailable during training, we use $w^{(t)}_{\text{tar}}(s, a)$ as its practical approximation: $\hat{w}^*_{\text{tar}}(s, a) \approx w^{(t)}_{\text{tar}}(s, a)$. To stabilize $\Delta\hat{w}^{(t)}_{\text{tar}}$, which can exhibit high variance due to limited target-domain data, we apply a moving average as follows:

$$\Delta\hat{w}^{(t)}_{\text{MA}} := \psi\Delta\hat{w}^{(t-1)}_{\text{MA}} + (1 - \psi)\Delta\hat{w}^{(t)}_{\text{tar}}, \tag{22}$$

where $\psi = 0.9$ is the smoothing factor. This ensures robust estimation of $\hat{\beta}(t)$, enhancing stability in the cross-domain policy extraction.

**Loss Function Optimization.** While $L_{\text{MAP}}$ and $L_{\text{BC}}$ mentioned in Section 3.2 yield solutions in the tabular setting, practical implementation constraints, such as scalability and computational efficiency, necessitate gradient-based updates using neural networks. In Algorithm 1, we train $G$, $H$, and the policy $\pi^{(t)}$ via stochastic gradient descent on $L_{\text{MAP}}$ and $L_{\text{BC}}$, respectively, leveraging neural network architectures to handle high-dimensional state-action spaces effectively.

## 4 Experiments

In this section, we present the main experimental results from the CDIL experiments, designed to evaluate the algorithms in benchmark robot control tasks. Our goal is to assess how effectively different methods leverage offline datasets from both source and target domains in policy learning. Unless stated otherwise, we report the average and the standard deviation over 5 random seeds for all the experimental results.

### 4.1 Setup

**Evaluation Domains.** We evaluate AdaptDICE on various continuous control tasks in MuJoCo and Robosuite to test the cross-domain transfer capability under varied dynamics, representations, and task complexities. Specifically:

- **MuJoCo**: We adopt standard MuJoCo environments, including Hopper-v3, HalfCheetah-v3, and Ant-v3, as the source domains. Following the procedures described in (Zhang et al., 2021), we modify these environments to construct the corresponding target domains.
- **Robosuite**: We utilize Robosuite (Zhu et al., 2020), a widely adopted framework for robot arm manipulation, to evaluate our algorithm on BlockLifting, DoorOpening and TableWiping tasks. For each task, the Panda robot arm serves as the source domain, while the UR5e robot arm is designated as the target domain.

The detailed specifications of the source-domain and target-domain morphologies are provided in Section D. The state and action space dimensions of each domain are provided in Table 5 in Section D.

**Baselines.** We compare AdaptDICE with recent CDIL algorithms, including SMODICE (Ma et al., 2022), GWIL (Fickinger et al., 2022), and the IL variant of IGDF (Wen et al., 2024). All the algorithms share the same source-domain and target-domain datasets for a fair comparison, with differences arising from the specific use of source-domain datasets, as detailed below:

Table 1: Configurations of the target-domain dataset in various scenarios.

| Environment | Dataset Set | Expert Traj. | Sub-optimal Traj. | Sub-optimal Composition |
|---|---|---|---|---|
| Hopper/TableWiping | Default | 1 | 60 | 10 expert + 50 random |
| Others * | | 1 | 101 | 1 expert + 100 random |
| Hopper/TableWiping | Expert Rich | 5 | 60 | 10 expert + 50 random |
| | Sub-Optimal Rich | 1 | 300 | 50 expert + 250 random |
| Others* | Expert Rich | 5 | 101 | 1 expert + 100 random |
| | Sub-Optimal Rich | 1 | 505 | 5 expert + 500 random |

* Others includes Ant, HalfCheetah, BlockLifting, and DoorOpening.

- **GWIL** (Fickinger et al., 2022): GWIL is an unsupervised CDIL method that learns from source-domain expert demonstrations as well as the target-domain data collected online. To adapt GWIL to the offline setting, we modify the online replay buffer to be supplied by the offline dataset. Due to the original experimental setup in the GWIL paper, we use only one source-domain expert trajectories in our experiments, without using source-domain imperfect demonstrations.

- **SMODICE** (Ma et al., 2022): SMODICE was originally designed to tackle imitation from dynamics or morphologically mismatched experts. To evaluate its performance in the Semi-Supervised CDIL setting, SMODICE can utilize the source-domain expert trajectories and the full target-domain dataset to enhance learning in the target domain, without using the source-domain imperfect demonstrations. This ensures a fair comparison without fundamentally changing the algorithm of SMODICE.

- **IGDF+IQLearn** (Wen et al., 2024): IGDF is a cross-domain representation learning algorithm that achieves data filtering via contrastive learning and can be integrated with any off-the-shelf RL method to address offline cross-domain RL. One can adapt IGDF to IL by using an off-the-shelf offline IL method, such as IQ-Learn (Garg et al., 2021). Specifically, the training can be done in two stages: (i) At the representation learning stage, IGDF learns to encode state-action representations that enable effective data filtering across domains. (ii) At the imitation stage, it leverages the learned encoders to select high-quality source trajectories for training, thereby improving the target-domain policy in CDIL.

**Dataset Configurations.** The target-domain datasets are constructed by mixing a small number of expert trajectories with a larger number of sub-optimal ones, following the configurations summarized in Table 1. Similarly, the source-domain datasets contain 400 expert trajectories and 2000 sub-optimal trajectories, where the sub-optimal set is composed of 400 expert and 1600 random rollouts. More detailed dataset specifications are provided in Section D.2.

## 4.2 Evaluation Results

In Table 2, we report the final performance values of AdaptDICE and baseline methods across various continuous control tasks, with the corresponding training curves provided in Figure 2 (Section D.3). AdaptDICE consistently outperforms the baseline CDIL algorithms in both MuJoCo and Robosuite, especially in high-dimensional robot arm manipulation tasks where learning from limited expert demonstrations is particularly challenging. We also observe that SMODICE and IGDF+IQ-Learn suffer on most of the tasks in this challenging low-data regime, and these results are consistent with those in their original papers, where hundreds of demonstrations are usually needed to achieve comparable return performance. Moreover, we observe that GWIL can barely make any learning progress. We hypothesize that this results from the fact that GWIL can only learn up to an isometry between the policies across domains due to its unsupervised nature. Recent works (Choi et al., 2023; Huang et al., 2024) also report that GWIL can struggle in practical cross-domain settings, suggesting that its need for expert supervision and inability to leverage imperfect demonstrations can limit its applicability.

Table 2: Experimental results of AdaptDICE and other baseline methods in the `Default` setting.

| Environment | AdaptDICE (Ours) | SMODICE | GWIL | IGDF+IQLearn |
|---|---|---|---|---|
| Hopper | $\mathbf{2289.9 \pm 328.7}$ | $2046.6 \pm 53.3$ | $9.8 \pm 0.03$ | $286.6 \pm 78.6$ |
| Ant | $\mathbf{3672.6 \pm 450.8}$ | $86.3 \pm 30.6$ | $-2891.6 \pm 17.4$ | $-358.3 \pm 1106.6$ |
| HalfCheetah | $\mathbf{4171.1 \pm 2659.2}$ | $-15.1 \pm 42.5$ | $-800.7 \pm 228.8$ | $2179.5 \pm 2422.1$ |
| BlockLifting | $\mathbf{38.5 \pm 13.9}$ | $0.8 \pm 0.6$ | $0.01 \pm 0.001$ | $7.3 \pm 5.4$ |
| DoorOpening | $\mathbf{139.5 \pm 30.8}$ | $9.6 \pm 14.6$ | $34.5 \pm 46.0$ | $70.0 \pm 52.8$ |
| TableWiping | $\mathbf{19.7 \pm 18.5}$ | $9.8 \pm 7.6$ | $0.03 \pm 0.6$ | $-15.5 \pm 34.9$ |

## 4.3 Ablation Study

To better understand the contributions of the source-domain and target-domain density ratios in AdaptDICE, we conduct an ablation study by using only the source-domain density ratio $w_{\mathrm{src}}$ or the target-domain density ratio $w_{\mathrm{tar}}$ during policy extraction. Specifically, we evaluate the following variants:

- AdaptDICE with $w_{\mathrm{src}}$ only: The policy is trained exclusively with the pre-trained source-domain density ratio, ignoring any target-domain density information. This is equivalent to choosing $\beta(t) = 1$, for all $t$.

- AdaptDICE with $w_{\mathrm{tar}}$ only: The policy relies solely on the target-domain density ratio learned from the offline target-domain data, without leveraging source-domain knowledge. This is equivalent to choosing $\beta(t) = 0$ for all $t$ and hence reduces to the single-domain DemoDICE (Kim et al., 2022).

Table 3: Experimental results of two ablation variants of AdaptDICE in the `Default` dataset setting.

| Environment | $w_{\mathrm{src}}$ only | $w_{\mathrm{tar}}$ only | AdaptDICE |
|---|---|---|---|
| Hopper | $2039.2 \pm 366.6$ | $\mathbf{2534.2 \pm 200.3}$ | $2289.9 \pm 328.7$ |
| Ant | $1047.8 \pm 129.1$ | $\mathbf{3921.5 \pm 247.1}$ | $3672.6 \pm 450.8$ |
| HalfCheetah | $-2.0 \pm 0.6$ | $-2.3 \pm 1.6$ | $\mathbf{4171.1 \pm 2659.2}$ |
| BlockLifting | $4.3 \pm 3.7$ | $11.9 \pm 13.3$ | $\mathbf{38.5 \pm 13.9}$ |
| DoorOpening | $60.1 \pm 16.5$ | $58.1 \pm 7.3$ | $\mathbf{139.5 \pm 30.8}$ |
| TableWiping | $2.9 \pm 4.1$ | $13.9 \pm 13.01$ | $\mathbf{19.7 \pm 18.5}$ |

In Table 3, we show the final performance values of AdaptDICE and its two ablation variants (*i.e.,* with $w_{\mathrm{src}}$ only and $w_{\mathrm{tar}}$ only), with the corresponding training curves shown in Figure 3 (Section D.3). In most of the tasks (HalfCheetah, BlockLifting, and DoorOpening), the full AdaptDICE method with source-to-target transfer significantly outperforms the variants using only $w_{\mathrm{src}}$ or $w_{\mathrm{tar}}$, highlighting the importance of combining source-domain and target-domain density ratios. In the Ant environment, the $w_{\mathrm{tar}}$-only variant of AdaptDICE performs well, while the $w_{\mathrm{src}}$-only variant performs rather poorly, suggesting the transfer mechanism remains rather reliable even under substantial discrepancy between the two domains.

## 4.4 Effect of Dataset Configurations

To evaluate AdaptDICE's reliability in various dataset configurations, we conduct additional experiments on the number of target-domain trajectories. As Semi-Supervised CDIL uses both labeled expert demonstrations $\mathcal{D}_{\mathrm{tar}}^{\mathrm{E}}$ and unlabeled or sub-optimal trajectories $\mathcal{D}_{\mathrm{tar}}^{\mathrm{I}}$, we vary the two dataset sizes separately: (i) increasing $|\mathcal{D}_{\mathrm{tar}}^{\mathrm{E}}|$ with fixed $|\mathcal{D}_{\mathrm{tar}}^{\mathrm{I}}|$ (`Expert Rich`) to assess annotation cost, and (ii) increasing $|\mathcal{D}_{\mathrm{tar}}^{\mathrm{I}}|$ with fixed $|\mathcal{D}_{\mathrm{tar}}^{\mathrm{E}}|$ (`Sub-Optimal Rich`) to evaluate robustness to imperfect data. The configurations are summarized in Table 1. This is meant to corroborate AdaptDICE's adaptability under the trade-off between supervision and sub-optimal trajectories in low-data regimes.

Table 4 shows that AdaptDICE reach higher final performance in both `Expert Rich` and `Sub-Optimal Rich` regimes than in the `Default` setting. In `Expert Rich`, additional labeled demonstrations accelerate convergence, while in `Sub-Optimal Rich`, abundant imperfect data improves representation and reduces

overfitting. These results demonstrate AdaptDICE's flexibility in trading off costly expert annotations for cheaper sub-optimal data, often yielding superior gains with a larger dataset size. We also report the training curves in Figure 4 (Section D.3).

Table 4: Experimental results of AdaptDICE under various dataset configurations.

| | MuJoCo Environments | | | Robosuite Environments | | |
|---|---|---|---|---|---|---|
| | **Hopper** | **Ant** | **HalfCheetah** | **BlockLifting** | **DoorOpening** | **TableWiping** |
| Default | $2289.9 \pm 328.7$ | $3672.6 \pm 450.8$ | $4171.1 \pm 2659.2$ | $38.5 \pm 13.9$ | $139.5 \pm 30.8$ | $19.7 \pm 18.4$ |
| Expert Rich | $\mathbf{2577.5 \pm 171.9}$ | $4494.2 \pm 760.1$ | $9296.0 \pm 1589.5$ | $65.1 \pm 17.3$ | $161.7 \pm 16.0$ | $37.6 \pm 18.9$ |
| Sub-Optimal Rich | $2381.3 \pm 299.3$ | $\mathbf{5038.1 \pm 207.1}$ | $\mathbf{10105.9 \pm 1821.7}$ | $\mathbf{68.9 \pm 25.7}$ | $\mathbf{181.4 \pm 5.0}$ | $\mathbf{44.1 \pm 20.0}$ |

## 5 Related Work

**Single-Domain Imitation Learning with Imperfect Demonstrations.** A central challenge in IL is ensuring robustness against noisy or suboptimal demonstrations. Early approaches extended behavioral cloning by explicitly modeling demonstration quality (Wu et al., 2019; Sasaki & Yamashina, 2021; Xu et al., 2022) or by re-weighting samples to place greater emphasis on higher-quality data (Tangkaratt et al., 2020; Wang et al., 2021). Other strategies employed adversarial training or regularization to mitigate the influence of low-quality data, such as through disagreement penalties (Brantley et al., 2019) or confidence-based filtering (Cao et al., 2022). Additional methods leveraged distribution matching and state-occupancy measures to stabilize learning from fixed, imperfect datasets, thereby addressing issues such as covariate shift and suboptimal actions (Brown et al., 2019; Kim et al., 2022; Ma et al., 2022; Yu et al., 2023; Mao et al., 2024). While these methods effectively account for demonstration quality, they typically assume that training and deployment data originate from the same domain. This assumption limits their applicability in scenarios requiring cross-domain transfer, a gap that CDIL is designed to address.

**Cross-Domain Imitation Learning under State and Action Discrepancies.** There are two major sub-categories in this setting:

- **CDIL with proxy tasks**: *Paired-trajectory* approaches exploit explicit correspondences between source and target domains, typically through observation or state alignment. Representative strategies include context translation (Raychaudhuri et al., 2021), time-contrastive networks (Sermanet et al., 2018), and state distribution matching (Liu et al., 2020). Extensions of these methods scale to multi-domain or long-horizon transfer, such as multi-domain behavioral cloning (Watahiki et al., 2024) and hierarchical skill alignment (Lin et al., 2024). Beyond visual or state-level alignment, several works incorporate skill-level decomposition to enhance generalization across tasks.

  In contrast, *unpaired-trajectory* methods do not rely on direct correspondences between source and target trajectories. Instead, they leverage adversarial objectives to facilitate transfer under limited supervision (Kim et al., 2020; Zolna et al., 2021; Giammarino et al., 2025). More recent efforts extend this line of work to handle unpaired data across variations in representation (Zhang et al., 2021) and dynamics (Kedia et al., 2025), thereby improving robustness in scenarios with heterogeneous domains.

- **Unsupervised CDIL**: Unsupervised approaches remove the need for paired data by aligning distributions or learning domain-invariant representations. Techniques such as optimal transport (Fickinger et al., 2022; Nguyen et al., 2021) and adversarial domain adaptation (Choi et al., 2023) mitigate discrepancies at the distributional or feature level, while embedding-based methods focus on extracting task-relevant representations for transfer (Franzmeyer et al., 2022; Pertsch et al., 2022). Recent advances further incorporate denoising strategies to enhance robustness against domain-induced noise (Huang et al., 2024).

**Cross-Domain Imitation Learning with Aligned State and Action Dimensions.** We categorize methods in this class as those where the source and target domains share identical state and action dimensions but differ in dynamics or contextual factors. These approaches can be broadly divided into two directions. First, *representation-level methods* aim to learn invariant features (Yin et al., 2022; Lyu et al., 2024a) or

contextual embeddings (Liu et al., 2023a) to enhance robustness across domains. For example, Wen et al. (2024) proposes a contrastive representation approach for data filtering in cross-domain offline reinforcement learning. We adopt and extend this approach as a baseline, incorporating dynamics-aware adjustments to better accommodate CDIL in heterogeneous settings. Second, *dynamics-adaptive methods* explicitly address discrepancies in transition dynamics, either by adapting policies to varying environments (Liu et al., 2023b) or by introducing off-dynamics objectives (Kang et al., 2024).

## 6 Conclusion

In this work, we proposed AdaptDICE, the first semi-supervised distribution-correction framework for Semi-Supervised CDIL that scales to heterogeneous state and action spaces. AdaptDICE integrates source-domain knowledge with limited offline target data without requiring paired trajectories or explicit domain mappings. We provided a theoretical analysis showing the convergence guarantee for density ratio estimation, and demonstrated through extensive experiments on MuJoCo and Robosuite that AdaptDICE consistently outperforms other baselines. Ablation studies further confirm the importance of combining source-domain and target-domain density ratios for stable transfer.

While AdaptDICE offers improved stability and flexibility in the Semi-Supervised CDIL setting, its effectiveness still depends on the availability of a small amount of labeled target data, which may not always be accessible in real-world scenarios. Future work includes exploring the integration of stronger representation learning or better offline adaptation mechanisms to further enhance cross-domain transferability.

## Acknowledgment

This research was partially supported by the National Science and Technology Council (NSTC) of Taiwan under Grant Numbers 114-2628-E-A49-002 and 114-2634-F-A49-002-MBK. This work was also partially supported by the Center for Intelligent Team Robotics and Human-Robot Collaboration under the "Top Research Centers in Taiwan Key Fields Program" of the Ministry of Education (MOE), Taiwan. We also thank the National Center for High-performance Computing (NCHC) for providing computational and storage resources.

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

## Appendices

## A  DemoDICE Algorithm

This section details the pseudo code of DemoDICE (Algorithm 2), an offline imitation learning method that trains a policy $\pi$ using expert ($\mathcal{D}^{\mathrm{E}}$) and imperfect ($\mathcal{D}^{\mathrm{I}}$) datasets, as formalized in Section 2.2. The algorithm proceeds in three stages: (1) pretraining a discriminator to derive a pseudo-reward $r(s,a)$; (2) updating a pseudo value network $\nu^{(t)}$ via gradient descent; and (3) optimizing a policy network $\pi^{(t)}$ through weighted behavior cloning. Below, we present the algorithm and explain each stage.

---

**Algorithm 2** DemoDICE

**Require:** Expert dataset $\mathcal{D}^{\mathrm{E}}$, imperfect dataset $\mathcal{D}^{\mathrm{I}}$, union dataset $\mathcal{D}^{\mathrm{U}} = \mathcal{D}^{\mathrm{E}} \cup \mathcal{D}^{\mathrm{I}}$

  1: Initialize discriminator network $c : \mathcal{S} \times \mathcal{A} \to [0,1]$

  2: Update $c \leftarrow \arg\min_{c:\mathcal{S}\times\mathcal{A}\to[0,1]} - \big( \mathbb{E}_{(s,a)\sim\mathcal{D}^{\mathrm{E}}}[\log c(s,a)] + \mathbb{E}_{(s,a)\sim\mathcal{D}^{\mathrm{U}}}[\log(1 - c(s,a))] \big)$

  3: Compute $r(s,a) = -\log(1/c(s,a) - 1)$ for $(s,a) \in \mathcal{D}^{\mathrm{U}}$

  4: Initialize critic network $\nu^{(0)}$, policy network $\pi^{(0)}$

  5: **for** iteration $t = 1, \ldots, T$ **do**

  6:      Sample batch $\mathcal{B}$ from $\mathcal{D}^{\mathrm{U}}$

  7:      Update $\nu^{(t)} \leftarrow \nu^{(t-1)} - \eta \nabla_\nu \tilde{L}(\nu^{(t-1)}; r, \mathcal{B})$

  8:      Update $\pi^{(t)} \leftarrow \arg\min_\pi \tilde{L}_{\mathrm{BC}}(\pi; \nu^{(t)}, \mathcal{B})$

  9: **return** Trained policy $\pi^{(T)}$

---

### Pretraining the Discriminator for Pseudo-Reward

The discriminator network $c : \mathcal{S} \times \mathcal{A} \to [0,1]$ is initialized and trained to distinguish expert state-action pairs from those in the union dataset $\mathcal{D}^{\mathrm{U}} = \mathcal{D}^{\mathrm{E}} \cup \mathcal{D}^{\mathrm{I}}$ by minimizing the binary cross-entropy loss: $J_c(\mathcal{D}^{\mathrm{E}}, \mathcal{D}^{\mathrm{U}}) := -\mathbb{E}_{(s,a)\sim\mathcal{D}^{\mathrm{E}}}[\log c(s,a)] - \mathbb{E}_{(s,a)\sim\mathcal{D}^{\mathrm{U}}}[\log(1 - c(s,a))]$. The resulting pseudo-reward is defined as $r(s,a) :=$

$-\log\left(1/c(s,a)-1\right)$. For the optimal discriminator $c^*(s,a) = \frac{d^{\mathrm{E}}(s,a)}{d^{\mathrm{E}}(s,a)+d^{\mathrm{U}}(s,a)}$, we have

$$-\log\left(\frac{1}{c^*(s,a)}-1\right) = \log\left(\frac{d^{\mathrm{E}}(s,a)}{d^{\mathrm{U}}(s,a)}\right), \tag{23}$$

which approximates the log-density ratio between expert and union distributions, providing a reward signal that emphasizes expert-like behaviors.

### Training the Critic Network

As described in Section 2.2, DemoDICE optimizes $\pi^* = \arg\max_\pi -D_{\mathrm{KL}}(d^\pi \parallel d^{\mathrm{E}}) - \alpha D_{\mathrm{KL}}(d^\pi \parallel d^{\mathrm{U}})$, yielding the dual problem with objective $L(\nu; r)$. The critic $\nu^{(t)}$ is updated for $t = 1, \dots, T$ by sampling a batch $\mathcal{B} \subset \mathcal{D}^{\mathrm{U}}$ and performing gradient descent:

$$\nu^{(t)} \leftarrow \nu^{(t-1)} - \eta\nabla_\nu\tilde{L}(\nu^{(t-1)}; r, \mathcal{B}), \tag{24}$$

where the empirical pseudo-value loss is defined as:

$$\tilde{L}(\nu^{(t-1)}; r, \mathcal{B}) := (1-\gamma)\mathbb{E}_{s\sim\mu}[\nu^{(t-1)}(s)] + (1+\alpha)\log\mathbb{E}_{(s,a)\sim\mathcal{B}}\left[\exp\left(\frac{A_{\nu^{(t-1)}}(s,a)}{1+\alpha}\right)\right], \tag{25}$$

with $A_{\nu^{(t-1)}}(s,a) := r(s,a) + \gamma\mathbb{E}_{s'\sim P(\cdot|s,a)}[\nu^{(t-1)}(s')] - \nu^{(t-1)}(s)$.

### Optimizing the Policy Network

The policy $\pi^{(t)}$ is updated by minimizing:

$$\tilde{L}_{\mathrm{BC}}(\pi; \nu^{(t)}, \mathcal{B}) := -\mathbb{E}_{(s,a)\sim\mathcal{B}}\left[\tilde{w}^{(t)}(s,a)\cdot\log\pi(a|s)\right], \tag{26}$$

where $\tilde{w}^{(t)}(s,a) := \exp(A_{\nu^{(t)}}(s,a)/(1+\alpha))$. This weighted behavior cloning loss, derived from the density ratio in Equation (3), aligns $\pi^{(t)}$ with the expert distribution over $T$ iterations, yielding $\pi^{(T)}$.

## B    Supporting Lemmas

In this section, we present the useful properties of the DemoDICE algorithm for the convergence analysis in the sequel. Recall that $\nu : \mathcal{S} \to \mathbb{R}$ denotes the state-dependent value function, $r(s,a) := \log\frac{d^{\mathrm{E}}(s,a)}{d^{\mathrm{U}}(s,a)}$ is the pseudo reward, the $\nu$-induced advantage function $A_\nu(s,a) := r(s,a) + \gamma\mathbb{E}_{s'\sim P(\cdot|s,a)}[\nu(s')] - \nu(s)$, $\mu \in \Delta(\mathcal{S})$ is the initial state distribution, $d^{\mathrm{U}} \in \Delta(\mathcal{S}\times\mathcal{A})$ is the underlying occupancy measure of the union dataset, $\alpha \geq 0$ is the KL-regularization coefficient, and $\lambda \geq 0$ is the $\ell_2$-regularization parameter. As described in Section 2.2, the training loss of DemoDICE is

$$L(\nu; r) := (1-\gamma)\mathbb{E}_{s\sim\mu}[\nu(s)] + (1+\alpha)\log\mathbb{E}_{(s,a)\sim d^{\mathrm{U}}}\left[\exp\left(\frac{A_\nu(s,a)}{1+\alpha}\right)\right]. \tag{27}$$

To simplify the notation, we use $L(\nu)$ as the shorthand of $L(\nu; r)$ in the sequel when the context is clear. For ease of exposition, we assume that both the state and action spaces are finite, and the results can be extended to continuous spaces with appropriate measure-theoretic adjustments. Let $S^* := \{\nu^* + C\cdot\mathbf{1}_\mathcal{S} \,|\, C \in \mathbb{R}\}$, where $\nu^*$ is a minimizer of $L$ and $\mathbf{1}_\mathcal{S}$ is the all-ones vector over $\mathcal{S}$ (Kim et al., 2022, Lemma 2). Denote the orthogonal complement of the span of $S^*$ in $\mathbb{R}^{|\mathcal{S}|}$ as $(S^*)^\perp$ and $\Pi_{S^*}(\nu) := \arg\min_{y\in S^*}\|\nu - y\|_2$, we assume that $L$ satisfies the quadratic growth on $\{\nu \,|\, \forall\nu \notin S^*\}$. *i.e.,* For all $\nu \notin S^*$ there exist a constant $c$ such that $L(\nu) - L^* \geq \frac{c}{2}\|\nu - \Pi_{S^*}(\nu)\|_2^2$.

**Lemma 1** (Properties of the DemoDICE Loss). *$L$ satisfies the following properties:*

*(a) $L$ is convex with respect to $\nu$.*

*(b) L is $L_f$-smooth, where $L_f = \frac{(1+\gamma)^2}{1+\alpha}$.*

*(c) $\nabla L(\nu)^\top \mathbf{1}_{\mathcal{S}} = 0$ for all $\nu \in \mathbb{R}^{|\mathcal{S}|}$.*

*Proof.* To establish (a), note that the convexity of $L$ with respect to $\nu$ follows directly from Proposition 2 in (Kim et al., 2022).

To establish (b), we start by decomposing the loss function as $L(\nu) = L_1(\nu) + L_2(\nu)$, where

$$L_1(\nu) := (1 - \gamma)\mathbb{E}_{s \sim \mu}[\nu(s)], \tag{28}$$

$$L_2(\nu) := (1 + \alpha) \log \mathbb{E}_{(s,a) \sim d^{\mathrm{U}}} \left[ \exp\left( \frac{A_\nu(s,a)}{1 + \alpha} \right) \right]. \tag{29}$$

We can check the Hessian $\nabla^2 L(\nu)$. Note that $\nabla^2 L_1(\nu) = 0$. As for the Hessian of $L_2(\nu)$, we first rewrite $L_2(\nu) = (1 + \alpha) \log g(\nu)$, where

$$g(\nu) := \mathbb{E}_{(s,a) \sim d^{\mathrm{U}}} \left[ \exp\left( \frac{A_\nu(s,a)}{1 + \alpha} \right) \right] = \sum_{(s,a)} d^{\mathrm{U}}(s,a) \exp\left( \frac{A_\nu(s,a)}{1 + \alpha} \right), \tag{30}$$

and define $h(\nu) := \log g(\nu)$. Then, we have $\nabla^2 L_2(\nu) = (1 + \alpha)\nabla^2 h(\nu)$. To simplify notation, let $b = \frac{1}{1+\alpha}$ and rewrite $A_\nu(s,a) = r(s,a) + l_{(s,a)}\nu$, where $l_{(s,a)} := \gamma P(\cdot|s,a) - \mathbf{e}_s \in \mathbb{R}^{|\mathcal{S}|}$ and $\mathbf{e}_s$ is the standard basis vector with 1 at state $s$. Moreover, define $w_{s,a} := d^{\mathrm{U}}(s,a) > 0$ and $x_{s,a} := b(r(s,a) + l_{s,a}\nu)$.

Then, we have

$$g(\nu) = \sum_{(s,a) \in \mathcal{S} \times \mathcal{A}} w_{s,a} \exp(x_{s,a}). \tag{31}$$

For each $s, a$, we define $p_{s,a} := w_{s,a} \exp(x_{s,a})/g(\nu)$, and $\{p_{s,a}\}$ forms a probability distribution over the state-action pairs. The Hessian of $h(\nu)$ can be derived as

$$\nabla^2 h(\nu) = b^2 \underbrace{\left[ \sum_{s,a} p_{s,a} l_{s,a}^\top l_{s,a} - \left( \sum_{s,a} p_{s,a} l_{s,a} \right)\left( \sum_{s,a} p_{s,a} l_{s,a} \right)^\top \right]}_{=:Q}. \tag{32}$$

To bound $Q$, we can compute its trace, *i.e.,*

$$\mathrm{trace}(Q) = \sum_{s,a} p_{s,a} \|l_{s,a}\|_2^2 - \left\| \sum_{s,a} p_{s,a} l_{s,a} \right\|_2^2 \leq \sum_{s,a} p_{s,a} \|l_{s,a}\|_2^2 \leq \max_{s,a} \|l_{s,a}\|_2^2, \tag{33}$$

where the last inequality follows from that $\sum_i p_i = 1$.

By Cauchy-Schwarz inequality, we have $Q \succeq 0$, and its largest eigenvalue satisfies $\lambda_{\max}(Q) \leq \mathrm{trace}(Q) \leq \max_{s,a} \|l_{s,a}\|_2^2$. This implies that

$$Q \preceq \left( \max_{s,a} \|l_{s,a}\|_2^2 \right) \mathbf{I}. \tag{34}$$

Moreover, we have

$$\|l_{s,a}\|_2^2 = \|\gamma P(\cdot|s,a) - \mathbf{e}_s\|_2^2 = \gamma^2 \sum_{s'} P(s'|s,a)^2 - 2\gamma P(s'|s,a) + 1. \tag{35}$$

Since $P(s'|s,a) \geq 0$ and $\sum_{s'} P(s'|s,a) = 1$, we have $\sum_{s'} P(s'|s,a)^2 \leq 1$ and $\|l_{s,a}\|_2^2 \leq \gamma^2 \cdot 1 + 2\gamma + 1 = (1+\gamma)^2$.

This also implies that

$$\nabla^2 L_2(\nu) \preceq \frac{(1+\gamma)^2}{1+\alpha}\mathbf{I}. \tag{36}$$

Therefore, $L$ is $L_f$-smooth with $L_f = \frac{(1+\gamma)^2}{1+\alpha}$.

To establish (c), we show that the gradient of $L$ is orthogonal to the all-ones vector $\mathbf{1}_\mathcal{S}$. Recall the decomposition $L(\nu) = L_1(\nu) + L_2(\nu)$ from the proof of (b):

$$L_1(\nu) = (1-\gamma)\mu^\top \nu, \tag{37}$$
$$L_2(\nu) = (1+\alpha)\log g(\nu), \tag{38}$$

where $g(\nu) = \sum_{(s,a)\in\mathcal{S}\times A} w_{s,a}\exp(x_{s,a})$, with $x_{s,a} = b(r(s,a) + l_{s,a}\nu)$, and $b = \frac{1}{1+\alpha}, l_{s,a} = \gamma P(\cdot|s,a) - \mathbf{e}_s$.

Then we have the gradient $\nabla L(\nu) = \nabla L_1(\nu) + \nabla L_2(\nu)$.

For $\nabla L_1(\nu)$, we have

$$\nabla L_1(\nu)^\top \mathbf{1}_\mathcal{S} = (1-\gamma)\mu^\top \mathbf{1}_\mathcal{S} = 1 - \gamma, \tag{39}$$

Since $\mu$ is a initial probability distribution.

For $\nabla L_2(\nu)$, we have $\nabla L_2(\nu) = (1+\alpha)\frac{\nabla g(\nu)}{g(\nu)}$, and inner sum is

$$\nabla L_2(\nu)^\top \mathbf{1}_\mathcal{S} = (1+\alpha)\frac{\nabla g(\nu)^\top \mathbf{1}_\mathcal{S}}{g(\nu)}. \tag{40}$$

Then we differentiating $g(\nu)$ gives $\frac{\partial g(\nu)}{\partial \nu(s')} = \sum_{(s,a)} w_{s,a}\exp(x_{s,a})\cdot b\cdot l_{s,a}(s')$, and

$$\nabla g(\nu)^\top \mathbf{1}_\mathcal{S} = \sum_{s'\in\mathcal{S}}\frac{\partial g(\nu)}{\partial \nu(s')} = b\sum_{(s,a)} w_{s,a}\exp(x_{s,a})\cdot\sum_{s'\in\mathcal{S}} l_{s,a}(s'). \tag{41}$$

Since the inner sum $\sum_{s'\in\mathcal{S}} l_{s,a}(s')$ in Equation (41) is

$$\sum_{s'\in\mathcal{S}} l_{s,a}(s') = \gamma\sum_{s'\in\mathcal{S}} P(s'|s,a) - 1 = \gamma - 1, \tag{42}$$

where $\sum_{s'\in\mathcal{S}} P(s'|s,a) = 1$.Therefore,

$$\nabla g(\nu)^\top \mathbf{1}_\mathcal{S} = b(\gamma-1)\sum_{s,a} w_{s,a}\exp(x_{s,a}) = b(\gamma-1)g(\nu). \tag{43}$$

Substituting back to Equation (40),

$$\nabla L_2(\nu)^\top \mathbf{1}_\mathcal{S} = (1+\alpha)\cdot\frac{b(\gamma-1)g(\nu)}{g(\nu)} = \gamma - 1. \tag{44}$$

Finally, we have

$$\nabla L(\nu)^\top \mathbf{1}_\mathcal{S} = \nabla L_1(\nu) + \nabla L_2(\nu) = (1-\gamma) + (\gamma-1) = 0. \tag{45}$$

The result holds for all $\nu \in \mathbb{R}^{|\mathcal{S}|}$. $\square$

**Lemma 2** (Convergence of DemoDICE). *Let $\nu^{(t)}$ and $w^{(t)}$ be the sequences generated under DemoDICE, with $|\mathcal{S}| < \infty$, and learning rate $\eta = 1/L_f$. Then:*

$$|w^{(t)}(s,a) - w^*(s,a)| \leq C_w w^*(s,a)\exp\left(\frac{C_w}{\sqrt{t}}\|\nu^{(0)} - \nu^*\|_2\right)\frac{1}{\sqrt{t}}\|\nu^{(0)} - \nu^*\|_2, \tag{46}$$

*where $\nu^* := \Pi_{\mathcal{S}^*}(\nu^{(0)})$, and $C_w := \frac{2(1+\gamma)\sqrt{L_f}}{\sqrt{c}(1+\alpha)}$.*

*Proof.* The proof begins by showing the convergence of the visitation distribution $\nu^{(t)}$ to the optimum $\nu^* := \Pi_{\mathcal{S}^*}(\nu^{(0)})$, followed by the convergence of the density ratio $w^{(t)}(s,a)$ to $w^*(s,a)$. Recall that $S^* := \{\nu^* + C \cdot \mathbf{1}_{\mathcal{S}} \,|\, C \in \mathbb{R}\}$ denotes the optimal level set of $L$. The projection $\Pi_{S^*}(\cdot)$ is taken with respect to the $\ell_2$-norm onto this affine subspace. Without loss of generality, we assume $\nu^{(0)}$ satisfies $\nu^{(0)} \notin S^*$. We start by establishing the convergence bound for $\|\nu^{(t)} - \nu^*\|^2$.

Since we assume $L$ is quadratic growth on $\{\nu \,|\, \forall \nu \notin S^*\}$ in the beginning of this section, and gradient descent with step size $\eta = 1/L_f$. By the result of Lemma 1(c), we have

$$\Pi_{S^*}(\nu^{(k)}) = \Pi_{S^*}(\nu^{(0)}) = \nu^*, \quad \forall k \geq 0. \tag{47}$$

Therefore, there exist a constant $c > 0$ such that for all $t \geq 0$

$$L(\nu^{(t)}) - L(\nu^*) \geq \frac{c}{2}\|\nu^{(t)} - \Pi_{S^*}(\nu^{(t)})\|_2^2 = \frac{c}{2}\|\nu^{(t)} - \nu^*\|_2^2. \tag{48}$$

Moreover, applying the result of (Bubeck et al., 2015, Theorem 3.3)

$$L(\nu^{(t)}) - L(\nu^*) \leq \frac{2L_f\|\nu^{(0)} - \nu^*\|_2^2}{t}, \tag{49}$$

we have

$$\|\nu^{(t)} - \nu^*\|_2^2 \leq \frac{2}{c}\left(L(\nu^{(t)}) - L(\nu^*)\right) \leq \frac{4L_f}{c} \cdot \frac{\|\nu^{(0)} - \nu^*\|_2^2}{t}, \tag{50}$$

which proves the first part.

Next, we establish the convergence of the density ratio $w^{(t)}(s,a)$ to $w^*(s,a)$. Their difference can be written as

$$w^{(t)}(s,a) - w^*(s,a) = \exp\left(\frac{A_{\nu^{(t)}}(s,a)}{1+\alpha}\right) - \exp\left(\frac{A_{\nu^*}(s,a)}{1+\alpha}\right). \tag{51}$$

Applying the Mean Value Theorem to the function $F(x) := \exp\left(\frac{x}{1+\alpha}\right)$, for each $(s,a)$ there exists $\xi^{(t)}(s,a) = \theta^{(t)}A_{\nu^{(t)}}(s,a) + (1-\theta^{(t)})A_{\nu^*}(s,a)$ for some $\theta^{(t)} \in [0,1]$ such that

$$\exp\left(\frac{A_{\nu^{(t)}}(s,a)}{1+\alpha}\right) - \exp\left(\frac{A_{\nu^*}(s,a)}{1+\alpha}\right) = \exp\left(\frac{\xi^{(t)}(s,a)}{1+\alpha}\right) \cdot \frac{A_{\nu^{(t)}}(s,a) - A_{\nu^*}(s,a)}{1+\alpha}. \tag{52}$$

Using $w^*(s,a) = \exp\left(A_{\nu^*}(s,a)/(1+\alpha)\right)$, we have

$$w^{(t)}(s,a) - w^*(s,a) = w^*(s,a) \cdot \exp\left(\frac{\xi^{(t)}(s,a) - A_{\nu^*}(s,a)}{1+\alpha}\right) \cdot \frac{A_{\nu^{(t)}}(s,a) - A_{\nu^*}(s,a)}{1+\alpha}. \tag{53}$$

To bound the difference $|A_{\nu^{(t)}}(s,a) - A_{\nu^*}(s,a)|$ in Equation (53):

$$|A_{\nu^{(t)}}(s,a) - A_{\nu^*}(s,a)| = \left|\gamma \sum_{s'} T(s'|s,a)(\nu^{(t)}(s') - \nu^*(s')) - (\nu^{(t)}(s) - \nu^*(s))\right| \leq (\gamma+1)\max_s|\nu^{(t)}(s) - \nu^*(s)|. \tag{54}$$

With the finite state space $|\mathcal{S}| < \infty$ and the established convergence of $\|\nu^{(t)} - \nu^*\|^2$, we have

$$\max_s|\nu^{(t)}(s) - \nu^*(s)| \leq \|\nu^{(t)} - \nu^*\|_2 \leq \sqrt{\frac{4L_f}{c \cdot t}\|\nu^{(0)} - \nu^*\|_2^2} = \frac{2\sqrt{L_f}}{\sqrt{c}} \frac{\|\nu^{(0)} - \nu^*\|_2}{\sqrt{t}}. \tag{55}$$

Therefore,

$$|A_{\nu^{(t)}}(s,a) - A_{\nu^*}(s,a)| \leq \frac{2(1+\gamma)\sqrt{L_f}}{\sqrt{c}}\frac{\|\nu^{(0)} - \nu^*\|_2}{\sqrt{t}}. \tag{56}$$

Next, we bound the exponential term $\exp\left(\frac{\xi^{(t)}(s,a)-A_{\nu^*}(s,a)}{1+\alpha}\right)$ in Equation (53). Since $\xi^{(t)}(s,a)$ is a convex combination of $A_{\nu^{(t)}}(s,a)$ and $A_{\nu^*}(s,a)$, we have

$$\left|\xi^{(t)}(s,a) - A_{\nu^*}(s,a)\right| \leq |A_{\nu^{(t)}}(s,a) - A_{\nu^*}(s,a)| \leq \frac{2(1+\gamma)\sqrt{L_f}}{\sqrt{c}}\frac{\|\nu^{(0)}-\nu^*\|_2}{\sqrt{t}}, \tag{57}$$

where the last inequality follows from Equation (56).

Consequently,

$$\exp\left(\frac{\xi^{(t)}(s,a) - A_{\nu^*}(s,a)}{1+\alpha}\right) \leq \exp\left(\frac{2(1+\gamma)\sqrt{L_f}}{\sqrt{c}(1+\alpha)}\frac{\|\nu^{(0)}-\nu^*\|_2}{\sqrt{t}}\right). \tag{58}$$

Substituting these bounds into Equation (53), we have

$$|w^{(t)}(s,a) - w^*(s,a)| \leq \frac{2w^*(s,a)(1+\gamma)\sqrt{L_f}}{\sqrt{c}(1+\alpha)}\exp\left(\frac{2(1+\gamma)\sqrt{L_f}}{\sqrt{c}(1+\alpha)}\frac{\|\nu^{(0)}-\nu^*\|_2}{\sqrt{t}}\right)\frac{1}{\sqrt{t}}\|\nu^{(0)}-\nu^*\|_2 \tag{59}$$

$$= C_w w^*(s,a)\exp\left(\frac{C_w}{\sqrt{t}}\|\nu^{(0)}-\nu^*\|_2\right)\frac{1}{\sqrt{t}}\|\nu^{(0)}-\nu^*\|_2, \tag{60}$$

where $C_w := \frac{2(1+\gamma)\sqrt{L_f}}{\sqrt{c}(1+\alpha)}$.

$\square$

## C   Convergence of AdaptDICE with Weighting Factor $\beta(t)$

In this section, we analyze the convergence of the AdaptDICE algorithm. Recall the definitions in Section 3.2, the cross-domain density ratio is given by $w_{\text{cross}}^{(t)}(s,a) = \beta(t)w_{\text{src}}(G^{(t)}(s), H^{(t)}(s,a)) + (1-\beta(t))w_{\text{tar}}^{(t)}(s,a)$, which combines a pre-trained source-domain density ratio and a target-domain density ratio through the weighting factor $\beta(t)$, where $\beta(t): \mathbb{N} \to [0,1]$ denotes a weighting factor that balances the contributions from the source and target domains. Where the mapping functions $G, H$ are optimized to minimize the cross-domain loss $\mathcal{L}_{\text{MAP}}$ Equation (4).

We further define the source and target domain error terms as follows:

$$\Delta w_{\text{src}}^{(t)}(s,a) := |w_{\text{src}}(G^{(t)}(s), H^{(t)}(s,a)) - w_{\text{tar}}^*(s,a)|, \quad \Delta w_{\text{tar}}^{(t)}(s,a) := |w_{\text{tar}}^{(t)}(s,a) - w_{\text{tar}}^*(s,a)|, \tag{61}$$

and

$$\Delta\bar{w}_{\text{src}}^{(t)} := \mathbb{E}_{(s,a)\sim\mathcal{D}_{\text{tar}}}|w_{\text{src}}(G^{(t)}(s), H^{(t)}(s,a)) - w_{\text{tar}}^*(s,a)|, \quad \Delta\bar{w}_{\text{tar}}^{(t)} := \mathbb{E}_{(s,a)\sim\mathcal{D}_{\text{tar}}}|w_{\text{tar}}^{(t)}(s,a) - w_{\text{tar}}^*(s,a)|, \tag{62}$$

To analyze convergence, we adopt the notations and assumptions from Section B, and define the optimal solution set of $\nu_{\text{tar}}$ as $S_{\text{tar}}^* := \{\nu_{\text{tar}}^* + C \cdot \mathbf{1}_{\mathcal{S}_{\text{tar}}} \mid C \in \mathbb{R}\}$ and orthogonal projection in target domain $\Pi_{S_{\text{tar}}^*} := \arg\min_{y\in S_{\text{tar}}^*}\|\nu_{\text{tar}} - y\|_2$.

**Theorem 1.** [Upper Bound of Cross-Domain Density Ratio Error Under AdaptDICE] *Under AdaptDICE, with learning rate $\eta \leq 1/L_f$, for each $(s,a)$, the cross-domain density ratio error is bounded as follows:*

$$|w_{cross}^{(t)}(s,a) - w_{tar}^*(s,a)| \leq \beta(t)\Delta w_{src}^{(t)}(s,a) + (1-\beta(t))\left[C_w w_{tar}^*(s,a)\exp\left(\frac{C_w}{\sqrt{t}}\|\nu_{tar}^{(0)}-\nu_{tar}^*\|_2\right)\frac{\|\nu_{tar}^{(0)}-\nu_{tar}^*\|_2}{\sqrt{t}}\right], \tag{16}$$

*where $L_f$ is the smoothness constant of $L_{DICE}$, $\nu_{tar}^* := \Pi_{S_{tar}^*}(\nu_{tar}^{(0)})$, and $C_w$ is a constant. Moreover, by selecting $\beta(t)$ as*

$$\beta(t) = \begin{cases} 0, & \text{if } \Delta\bar{w}_{tar}^{(t)} \leq \Delta\bar{w}_{src}^{(t)} \\ 1, & \text{otherwise} \end{cases}$$

*the expected cross-domain density ratio error satisfies*

$$\mathbb{E}_{(s,a)\sim\mathcal{D}_{tar}}|w_{cross}^{(t)}(s,a) - w_{tar}^*(s,a)| \leq \min(\Delta\bar{w}_{src}^{(t)}, \Delta\bar{w}_{tar}^{(t)}). \tag{17}$$

*Proof.* The cross-domain density ratio error is bounded using the triangle inequality:

$$|w_{cross}^{(t)}(s,a) - w_{tar}^*(s,a)| \leq \beta(t)|w_{src}(G^{(t)}(s), H^{(t)}(s,a)) - w_{tar}^*(s,a)| + (1-\beta(t))|w_{tar}^{(t)}(s,a) - w_{tar}^*(s,a)| \tag{63}$$

$$= \beta(t)\Delta w_{src}^{(t)}(s,a) + (1-\beta(t))\Delta w_{tar}^{(t)}(s,a). \tag{64}$$

For the second term in Equation (64), we bound $\Delta w_{tar}^{(t)}(s,a)$. It follows from Lemma 2 that:

$$|w_{tar}^{(t)}(s,a) - w_{tar}^*(s,a)| \leq C_w w_{tar}^*(s,a) \exp\left(\frac{C_w}{\sqrt{t}}\|\nu_{tar}^{(0)} - \nu_{tar}^*\|_2\right)\frac{1}{\sqrt{t}}\|\nu_{tar}^{(0)} - \nu_{tar}^*\|_2, \tag{65}$$

where $C_w = \frac{2(1+\gamma)\sqrt{L_f}}{\sqrt{c}(1+\alpha)}$.

We then combine the bounds using $\beta(t)$. Substitute Equation (65) into Equation (64):

$$|w_{cross}^{(t)}(s,a) - w_{tar}^*(s,a)| \leq \beta(t)\cdot\Delta w_{src}^{(t)}(s,a) + (1-\beta(t))\cdot\left[C_w w_{tar}^*(s,a)\exp\left(\frac{C_w}{\sqrt{t}}\|\nu_{tar}^{(0)} - \nu_{tar}^*\|_2\right)\frac{\|\nu_{tar}^{(0)} - \nu_{tar}^*\|_2}{\sqrt{t}}\right], \tag{66}$$

Having established the pointwise convergence bounds for both $\lambda > 0$ and $\lambda = 0$, we next show that the expected error satisfies $\mathbb{E}_{(s,a)\sim\mathcal{D}_{tar}}|w_{cross}^{(t)}(s,a) - w_{tar}^*(s,a)| \leq \min(\Delta\bar{w}_{src}^{(t)}, \Delta\bar{w}_{tar}^{(t)})$. Taking the expectation over Equation (64), we have

$$\mathbb{E}_{(s,a)\sim\mathcal{D}_{tar}}\left[|w_{cross}^{(t)}(s,a) - w_{tar}^*(s,a)|\right] \leq \mathbb{E}_{(s,a)\sim\mathcal{D}_{tar}}\left[\beta(t)\Delta w_{src}^{(t)}(s,a) + (1-\beta(t))\Delta w_{tar}^{(t)}(s,a)\right]. \tag{67}$$

By linearity of expectation, this becomes:

$$\beta(t)\mathbb{E}_{(s,a)\sim\mathcal{D}_{tar}}[\Delta w_{src}^{(t)}(s,a)] + (1-\beta(t))\mathbb{E}_{(s,a)\sim\mathcal{D}_{tar}}[\Delta w_{tar}^{(t)}(s,a)] = \beta(t)\Delta\bar{w}_{src}^{(t)} + (1-\beta(t))\Delta\bar{w}_{tar}^{(t)}. \tag{68}$$

Choosing $\beta(t) = \mathbb{I}\left[\Delta\bar{w}_{src}^{(t)} < \Delta\bar{w}_{tar}^{(t)}\right]$, we obtain:

- If $\Delta\bar{w}_{src}^{(t)} \geq \Delta\bar{w}_{tar}^{(t)}$, then $\beta(t) = 0$, and the expected error is $\Delta\bar{w}_{tar}^{(t)} = \min(\Delta\bar{w}_{src}^{(t)}, \Delta\bar{w}_{tar}^{(t)})$.

- If $\Delta\bar{w}_{src}^{(t)} < \Delta\bar{w}_{tar}^{(t)}$, then $\beta(t) = 1$, and the expected error is $\Delta\bar{w}_{src}^{(t)} = \min(\Delta\bar{w}_{src}^{(t)}, \Delta\bar{w}_{tar}^{(t)})$.

Thus, the expected error satisfies $\mathbb{E}_{(s,a)\sim\mathcal{D}_{tar}}|w_{cross}^{(t)}(s,a) - w_{tar}^*(s,a)| \leq \min(\Delta\bar{w}_{src}^{(t)}, \Delta\bar{w}_{tar}^{(t)})$.

Furthermore, when $\Delta\bar{w}_{src}^{(t)} < \Delta\bar{w}_{tar}^{(t)}$, selecting $\beta(t) = 1$ yields an expected error bound of $\Delta\bar{w}_{src}^{(t)}$, which is smaller than $\Delta\bar{w}_{tar}^{(t)}$, the error obtained by training solely in the target domain with DemoDICE. This implies a faster convergence rate for AdaptDICE in such cases.

$\square$

# D Detailed Experimental Setup

## D.1 Settings of Evaluation Environments

To evaluate our proposed AdaptDICE framework, inspired by (Pan et al., 2024; Chen et al., 2026), we conduct experiments in MuJoCo (Todorov et al., 2012) and Robosuite (Zhu et al., 2020), two physics-based simulation frameworks widely used for reinforcement and imitation learning. MuJoCo offers locomotion tasks with realistic dynamics, while Robosuite provides robotic manipulation tasks, together serving as complementary benchmarks for testing offline cross-domain imitation learning (CDIL) algorithms.

To systematically examine AdaptDICE under varied cross-domain discrepancies, we design source–target environment pairs that differ in morphology, observation space, and control dimensionality, while maintaining identical task objectives. For MuJoCo tasks, the domain gap is introduced by modifying the agent's embodiment (e.g., adding legs or joints), which alters both the dynamics and state–action representations.

**Hopper with an extra thigh:** We add an extra thigh body attached to the torso. following analogous modifications used in (Xu et al., 2023). Detailed modifications of the XML file are:

```
1  <body name="thigh1" pos="0 0 1.45">
2      <joint axis="0 -1 0" name="thigh_joint1" pos="0 0 1.45" range="-150 0" type="hinge"/>
3      <geom friction="0.9" fromto="0 0 1.45 0 0 1.85" name="thigh_geom1" size="0.05" type="capsule"/>
4  </body>
```

**Ant with an extra leg:** We add a fifth leg ("middle_leg") to the central body, inspired by the cross-domain transfer settings in (Zhu et al., 2024), which similarly modifies agent morphology to induce representation and dynamics gaps. Detailed modifications of the XML file are:

```
1  <body name="middle_leg" pos="0 0 0">
2      <geom fromto="0.0 0.0 0.0 0.0 -0.28 0.0" name="aux_5_geom" size="0.08" type="capsule"/>
3      <body name="aux_5" pos="0.0 -0.28 0.0">
4          <joint axis="0 0 1" name="hip_5" pos="0.0 0.0 0.0" range="-10 10" type="hinge"/>
5          <geom fromto="0.0 0.0 0.0 0.0 -0.28 0.0" name="middle_leg_geom" size="0.08"
           ↪   type="capsule"/>
6          <body pos="0.0 -0.28 0.0">
7              <joint axis="1 1 0" name="ankle_5" pos="0.0 0.0 0.0" range="30 70" type="hinge"/>
8              <geom fromto="0.0 0.0 0.0 0.0 -0.56 0.0" name="fifth_ankle_geom" size="0.08"
               ↪   type="capsule"/>
9          </body>
10      </body>
11 </body>
```

**HalfCheetah with an extra back leg:** We follow the morphological modification introduced by (Zhang et al., 2021), adding a second back leg set (thigh, shin, foot) to create embodiment mismatches while preserving the original locomotion objective. Detailed modifications of the XML file are:

```
1  <body name="bthigh2" pos="-.5 .06 0">
2      <joint axis="0 1 0" damping="6" name="bthigh2" pos="0 0 0" range="-.52 1.05" stiffness="240"
       ↪   type="hinge"/>
3      <geom axisangle="1 -1 0 3.8" name="bthigh2" pos=".1 0 -.13" size="0.046 .145" type="capsule"/>
4      <body name="bshin2" pos=".16 .06 -.25">
5          <joint axis="0 1 0" damping="4.5" name="bshin2" pos="0 0 0" range="-.785 .785"
           ↪   stiffness="180" type="hinge"/>
6          <geom axisangle="0 1 0 -2.03" name="bshin2" pos="-.14 0 -.07" rgba="0.9 0.6 0.6 1"
           ↪   size="0.046 .15" type="capsule"/>
7          <body name="bfoot2" pos="-.28 0 -.14">
8              <joint axis="0 1 0" damping="3" name="bfoot2" pos="0 0 0" range="-.4 .785"
               ↪   stiffness="120" type="hinge"/>
9              <geom axisangle="0 1 0 -.27" name="bfoot2" pos=".03 0 -.097" rgba="0.9 0.6 0.6 1"
               ↪   size="0.046 .094" type="capsule"/>
```

```
10            </body>
11         </body>
12  </body>
```

For Robosuite tasks, we evaluate transfer across robot arms (Panda → UR5e) performing identical manipulation tasks, introducing embodiment and kinematic mismatches while keeping object configurations and goals consistent. This ensures that performance differences arise from cross-domain transfer capability rather than task definition. The corresponding state and action dimensions of each source–target pair are summarized in Table 5.

Table 5: State and action dimensions of the source and target domains.

| Environment | Source | Target |
|---|---|---|
| | State/Action | State/Action |
| Hopper | 11 / 3 | 13 / 4 |
| HalfCheetah | 17 / 6 | 23 / 9 |
| Ant | 27 / 8 | 31 /10 |
| BlockLifting | 42 / 8 | 47 / 7 |
| DoorOpening | 46 / 8 | 51 / 7 |
| TableWiping | 37 / 7 | 34 / 6 |

Below, we present the source–target environment pairs used in our experiments. For each task, the left image shows the source domain, and the right image shows the corresponding target domain.

- **Hopper and Three-Thigh Hopper**:

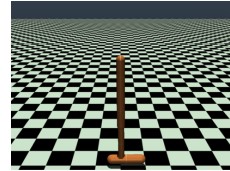 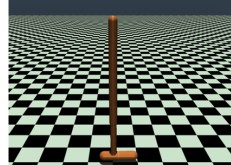

- **Ant and Five-Leg Ant**:

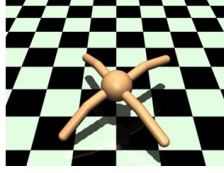 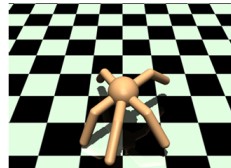

- **HalfCheetah and Three-Leg HalfCheetah**:

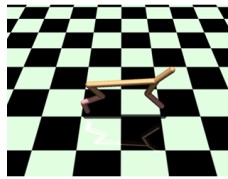 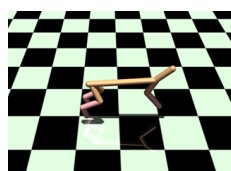

- **Panda and UR5e BlockLifting**:

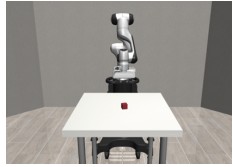 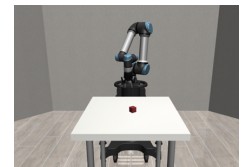

- **Panda and UR5e DoorOpening**:

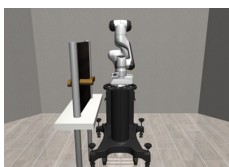 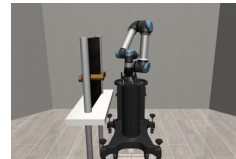

- **Panda and UR5e TableWiping**:

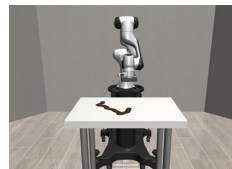 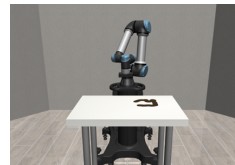

### D.2 Dataset Configuration of Each Experiment

Table 6 summarizes the target-domain dataset configurations used in our experiments. Each dataset is constructed by mixing a small number of expert trajectories with a larger set of sub-optimal trajectories, which consist of expert demonstrations and purely random rollouts. We categorize the datasets based on either the proportion of expert versus sub-optimal trajectories (Default, Expert Rich, Sub-Optimal Rich) or the overall scale of available data (Data Rich). For all experiments, we used the same dataset setting of source domain, as described in Table 7.

Table 6: Configurations (including `Data Rich`) of the target-domain dataset in various scenarios.

| Environment | Dataset Set | Expert Traj. | Sub-optimal Traj. | Sub-optimal Composition |
|---|---|---|---|---|
| Hopper/TableWiping | Default | 1 | 60 | 10 expert + 50 random |
| Others* | | 1 | 101 | 1 expert + 100 random |
| Hopper/TableWiping | Expert Rich | 5 | 60 | 10 expert + 50 random |
| | Sub-Optimal Rich | 1 | 300 | 50 expert + 250 random |
| Others* | Expert Rich | 5 | 101 | 1 expert + 100 random |
| | Sub-Optimal Rich | 1 | 505 | 5 expert + 500 random |
| All | Data Rich | 10 | 800 | 400 expert + 400 random |

* Others includes Ant, HalfCheetah, BlockLifting, and DoorOpening.

Table 7: Configuration of the source-domain dataset.

| Expert Traj. | Sub-optimal Traj. | Sub-optimal Composition |
|---|---|---|
| 400 | 2000 | 400 expert + 1600 random |

Regarding the acquisition of expert demonstrations, in addition to the official MuJoCo `.hdf5` datasets provided by D4RL (Fu et al., 2020), the remaining expert datasets were generated by training policies with Soft Actor-Critic (SAC) (Haarnoja et al., 2018). Specifically, we train an SAC agent until it reaches expert-level performance, and then use the trained policy to collect expert trajectories.

## D.3 Training Curves

We provide the training curves of evaluation, ablation study, and data configuration experiments described in Section 4.

**Evaluation Results.** The experimental results in Figure 2 demonstrate that AdaptDICE achieves the superior and stable performance across all testing environments. Compared to baseline methods such as SMODICE, GWIL, and IGDF+IQ-Learn, AdaptDICE not only exhibits faster convergence during the early stages of training but also attains significantly higher final average returns; while other methods largely struggle to learn effective policies in these settings, AdaptDICE maintains robust performance.

**Ablation Study.** Based on the results in Figure 3, our ablation study confirms the criticality of combining both source and target domain weights. While the Target-Only variant suffers from severe performance collapse in environments such as HalfCheetah, the Source-Only variant yields low returns due to limited transferability. However, despite the poor performance of the Source-Only baseline, it is essential for the stability of our method. The proposed hybrid density $w_{\text{cross}}$ effectively leverages this source information to anchor the learning process, ensuring that AdaptDICE maintains robust performance in the later stages of training without diverging.

**Effect of Dataset Configurations.** The results in Figure 4 indicate that expanding the dataset scale enhances model performance; both increasing expert data (`Expert-Rich`) and sub-optimal data (`Sub-Optimal Rich`) yield improvements over the `Default` configuration. Notably, utilizing a large amount of sub-optimal data (indicated by the blue line) achieves the best performance in most tasks. This demonstrates Adapt-DICE's strong capability to extract useful information from imperfect demonstrations to refine its policy.

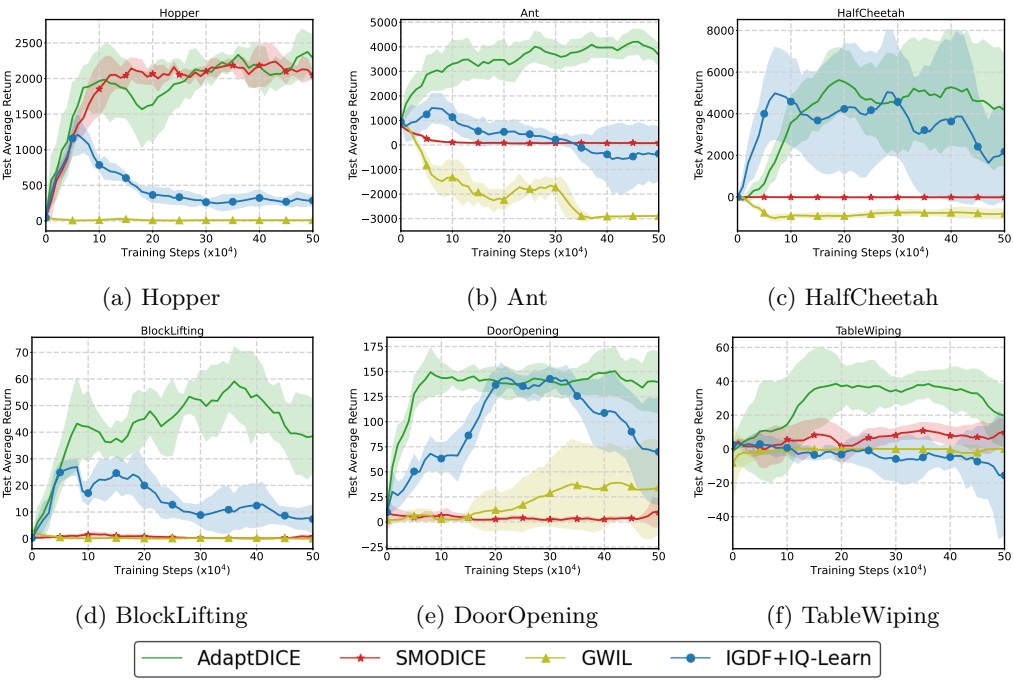

Figure 2: Training curves of AdaptDICE and the baseline methods in the `Default` setting: (a)-(c) MuJoCo locomotion tasks; (d)-(f) Robot arm manipulation tasks in Robosuite.

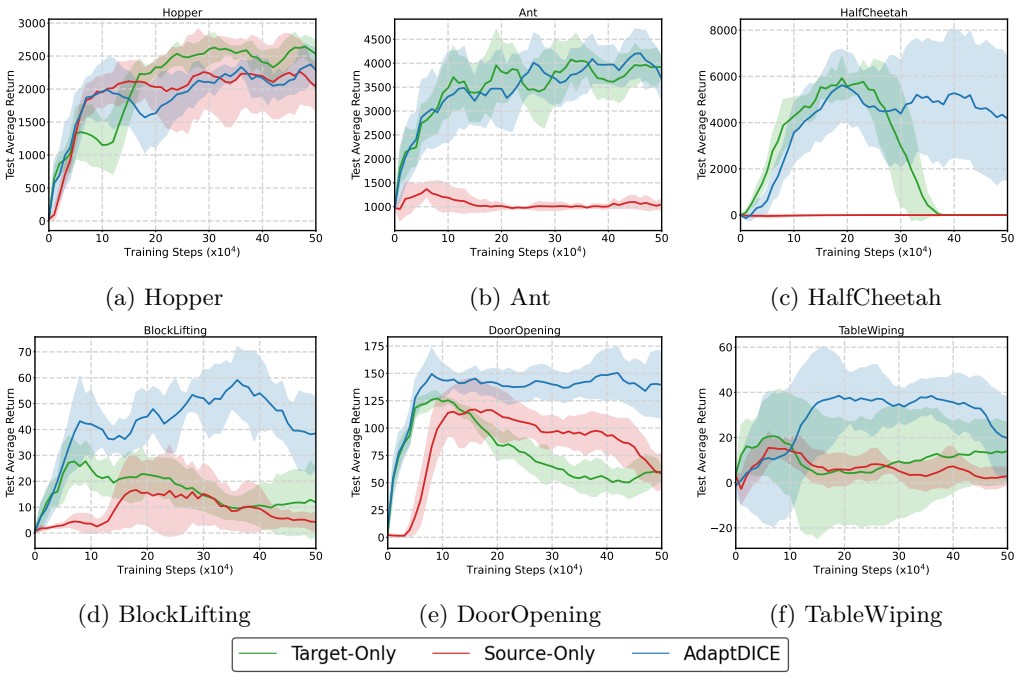

Figure 3: Ablation study: Training curves of the full AdaptDICE and its two ablation variants (*i.e.,* using only $w_{\mathrm{src}}$ or $w_{\mathrm{tar}}$).

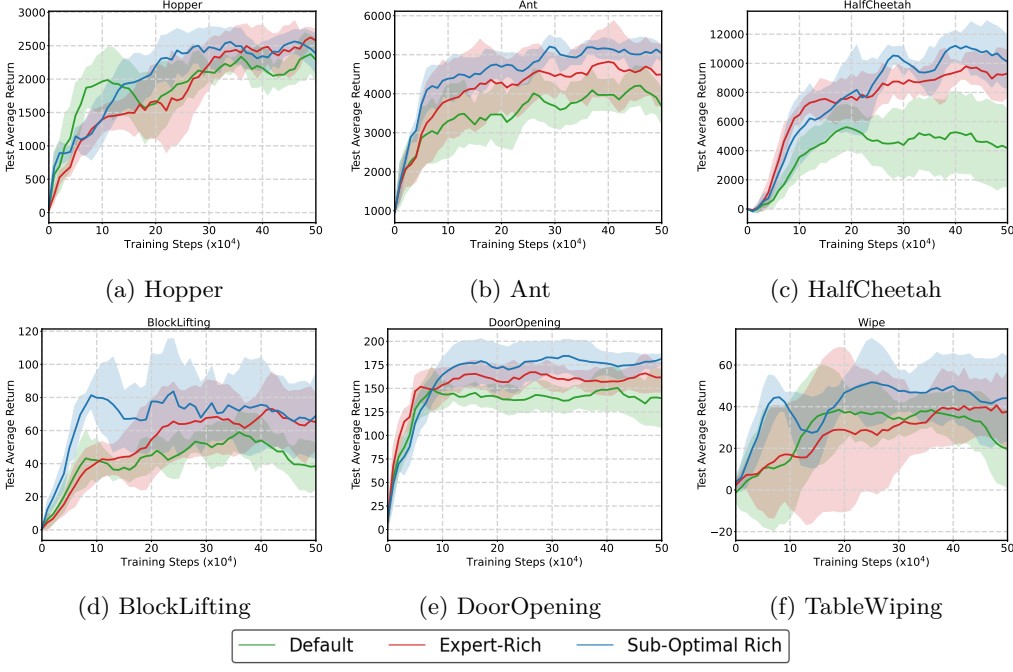

Figure 4: Effect of dataset configurations under AdaptDICE: We compare performance across the `Expert Rich` and `Sub-Optimal Rich` regimes, showing that both types of data expansion lead to improvements over the `Default` dataset setting.

## D.4 Effect of Dataset Configurations on Baseline Methods

In the main text, we presented the analysis of AdaptDICE under varying dataset configurations, highlighting that both `Expert Rich` and `Sub-Optimal Rich` regimes improve performance compared to the `Default`. To assess whether other baseline methods (SMODICE, GWIL, and IGDF+IQLearn) also benefit from increased data quantities, we conduct similar experiments for these algorithms, using the `Data Rich` configurations outlined in Table 6.

The training curves in Figures 5 to 7, as well as those final performance values in Table 8 and Table 9, illustrate the impact of increased dataset sizes on the performance of SMODICE, GWIL, and IGDF+IQ-Learn across the evaluated environments. Compared to the `Default` settings, SMODICE and IGDF+IQ-Learn exhibit consistent performance improvements under the `Data Rich` settings. In contrast, GWIL shows limited improvement under `Data Rich` settings, with consistently poor performance across all environments. This can be attributed to GWIL's reliance on online learning and its unsupervised Cross-Domain Imitation Learning (CDIL) assumption of isomorphic state-action spaces. The isomorphic assumption often fails in diverse environments with varying dynamics, leading to poor generalization. Moreover, GWIL's online learning nature limits its ability to effectively utilize larger offline datasets, as it struggles to adapt to increased data variety without explicit supervision or alignment with target domain dynamics.

These results highlight that methods like SMODICE and IGDF+IQ-Learn, which are designed to handle offline data efficiently, benefit significantly from increased dataset sizes, while GWIL's performance is constrained by its methodological assumptions and online learning requirements.

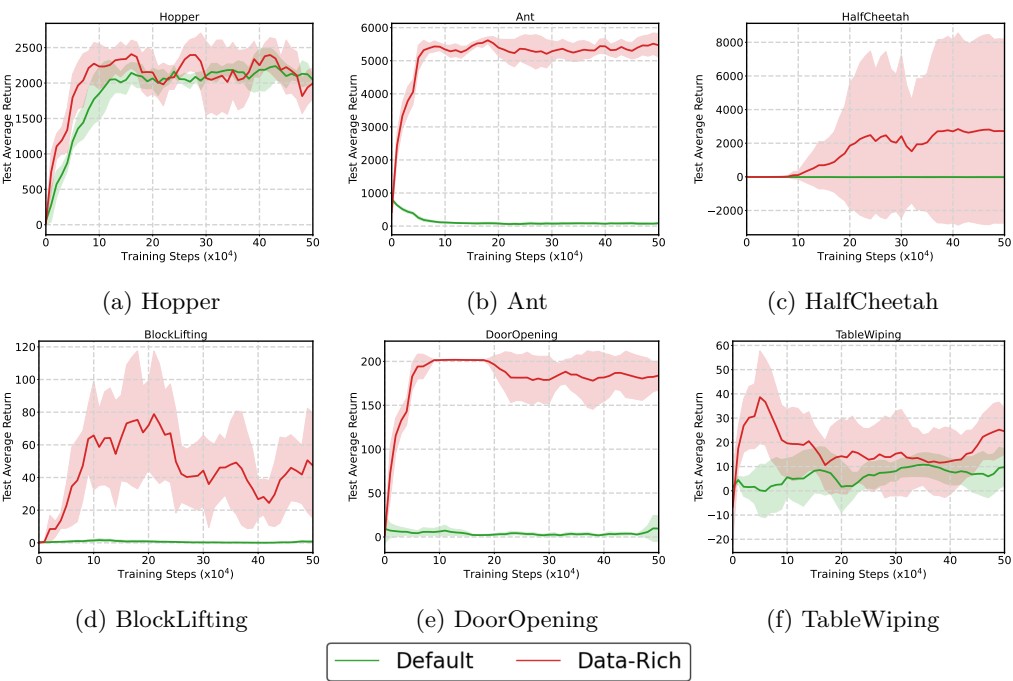

Figure 5: Effect of dataset configurations under SMODICE.

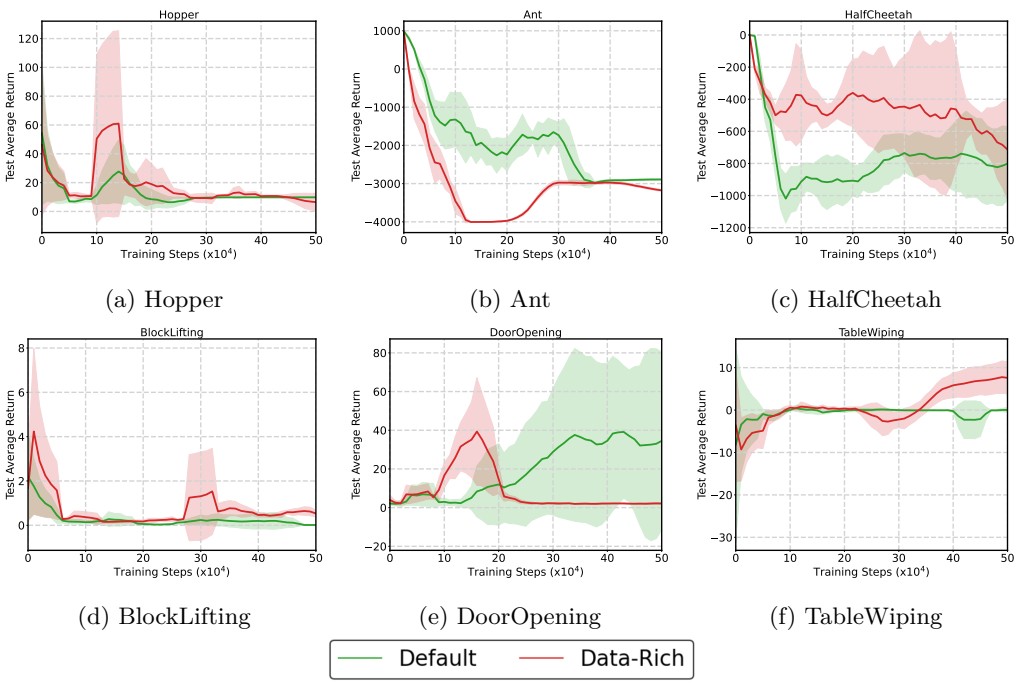

Figure 6: Effect of dataset configurations under GWIL.

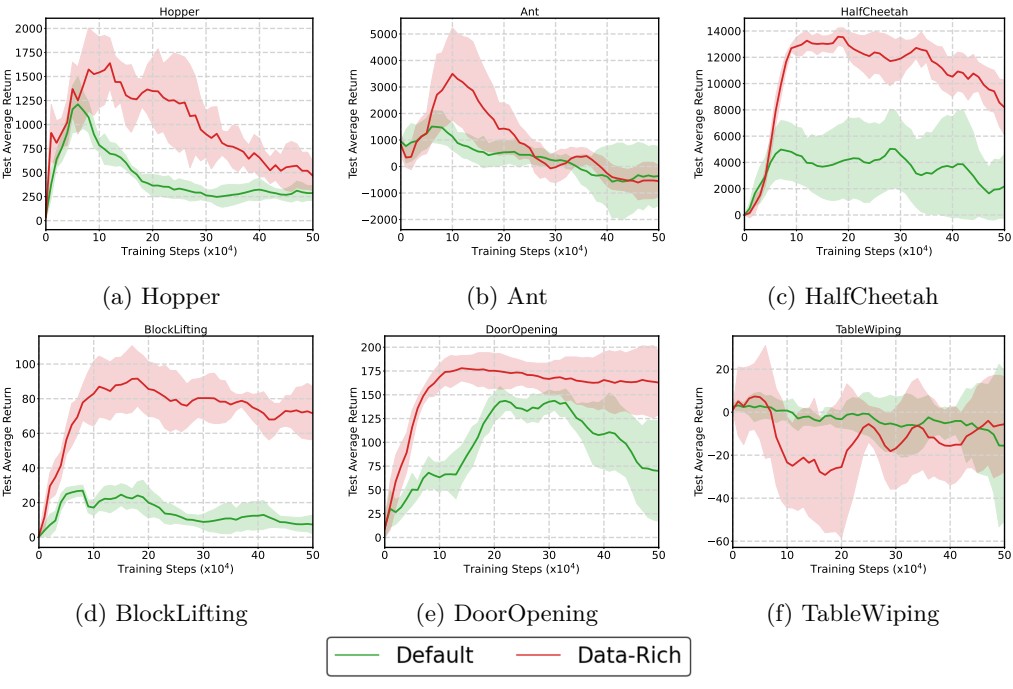

Figure 7: Effect of dataset configurations under IGDF+IQ-Learn.

Table 8: Experimental results of other baseline methods in `Data Rich` setting.

| Environment | SMODICE | GWIL | IGDF+IQLearn |
|---|---|---|---|
| Hopper | $1996.0 \pm 219.1$ | $6.4 \pm 6.1$ | $466.3 \pm 153.3$ |
| Ant | $5467.9 \pm 2673.1$ | $-3177.1 \pm 1559.0$ | $-551.6 \pm 582.7$ |
| HalfCheetah | $2718.7 \pm 5468.2$ | $-713.8 \pm 307.8$ | $8184.5 \pm 2083.3$ |
| BlockLifting | $47.3 \pm 32.2$ | $0.5 \pm 0.1$ | $71.5 \pm 15.3$ |
| DoorOpening | $184.0 \pm 16.6$ | $2.1 \pm 0.3$ | $162.8 \pm 33.6$ |
| TableWiping | $24.6 \pm 9.5$ | $7.6 \pm 3.8$ | $-5.6 \pm 22.1$ |

Table 9: Experimental results of other baseline methods in `Default` setting.

| Environment | SMODICE | GWIL | IGDF+IQLearn |
|---|---|---|---|
| Hopper | $2046.6 \pm 53.3$ | $9.8 \pm 0.03$ | $286.6 \pm 78.6$ |
| Ant | $86.3 \pm 30.6$ | $-2891.6 \pm 17.4$ | $-358.3 \pm 1106.6$ |
| HalfCheetah | $-15.1 \pm 42.5$ | $-800.7 \pm 228.8$ | $2179.5 \pm 2422.1$ |
| BlockLifting | $0.8 \pm 0.6$ | $0.01 \pm 0.001$ | $7.3 \pm 5.4$ |
| DoorOpening | $9.6 \pm 14.6$ | $34.5 \pm 46.0$ | $70.0 \pm 52.8$ |
| TableWiping | $9.8 \pm 7.6$ | $0.03 \pm 0.6$ | $-15.5 \pm 34.9$ |

### D.5 Computational Resources and Hyperparameters

All experiments were conducted on a high-performance computing cluster. The training jobs were executed on a server equipped with an Intel Xeon Gold 6154 CPU (3.0 GHz, 36 cores) and eight NVIDIA Tesla V100 GPUs, each with 32 GB of memory. The software environment was based on CUDA version 12.9 to ensure compatibility with the GPU architecture. This computational setup provided sufficient resources to support large-batch training and efficient parallelization across multiple GPUs, which was essential for handling the complexity of our experiments.

Table 10: Hyperparameters used in our experiments.

| Component | Setting |
|---|---|
| $\gamma$ (discount factor) | 0.99 |
| $\alpha$ (regularization coefficient) | 0.05 |
| Policy hidden dim | (256, 256) |
| Critic hidden dim | (256, 256) |
| Cost hidden dim | (256, 256) |
| Batch size | 512 |
| Optimizer | Adam |
| Learning rate | $\{3 \times 10^{-4}, 1 \times 10^{-4}\}$ |
| Gradient penalty coeffs | $(0.1, 1 \times 10^{-4})$ |
| Actor L2 penalty | $1 \times 10^{-2}$ |
| Mapping networks (decoder / action decoder) | (256, 256), Adam, $3 \times 10^{-4}$ |

### D.6 Implementation of Baseline Methods

In this subsection, we provide the detailed implementation of the baseline methods considered in our experiments as follows:

**DemoDICE.** DemoDICE (Kim et al., 2022) is an offline imitation learning algorithm that directly optimizes a density ratio estimator to recover the pseudo reward function. It has been shown to be effective in settings with limited demonstrations by exploiting unlabeled trajectories. We used the official PyTorch implementation from `https://github.com/geon-hyeong/imitation-dice`.

**SMODICE.** SMODICE (Ma et al., 2022) is an offline imitation learning algorithm that performs state-occupancy matching via distribution correction estimation, suitable for cross-domain transfers. We used the official PyTorch implementation from `https://github.com/JasonMa2016/SMODICE`. To adapt SMODICE for our CDIL experiments, we trained the discriminator using expert observations from source domain and offline dataset from target domain to estimate state occupancy distributions. The policy was trained using target-domain imperfect demonstrations, as these provide diverse behavior for robust learning. Contrary to the original paper's assumption of identical state-action spaces, we applied zero-padding or truncation to align state-action vectors and ensure dimensional consistency.

**GWIL.** GWIL (Fickinger et al., 2022) is a cross-domain imitation learning method that employs Gromov-Wasserstein optimal transport to align behavioral distributions between source (expert) and target (learner) domains, enabling transfer without requiring paired data or dimension matching. We used the official PyTorch implementation from `https://github.com/facebookresearch/gwil`. Originally designed for online settings, we adapted GWIL to offline CDIL by replacing the online replay buffer with an offline dataset of imperfect demonstrations. Consistent with the original paper, which uses only 1 source demonstration for training, we also employed 1 source domain expert trajectory in our experiments

**IGDF+IQ-Learn.** IGDF (Wen et al., 2024) is a cross-domain offline reinforcement learning method that uses contrastive learning to learn invariant state-action representations for data filtering across domains. We adapted it to offline imitation learning (IL) by integrating IQ-Learn (Garg et al., 2021), which optimizes policies directly from demonstrations without adversarial training. We used the official PyTorch implementations from `https://github.com/BattleWen/IGDF` (IGDF) and `https://github.com/Div-Infinity/IQ-Learn` (IQ-Learn). In the encoder phase, we trained a contrastive encoder using imperfect demonstrations from both source and target domains. In the IL phase, we filtered high-quality source domain expert trajectories ($\xi = 0.5$, selecting the top-50% based on representation similarity) and used target domain expert demonstrations to train the policy via IQ-Learn. Despite the paper's assumption of identical state-action spaces, we applied zero-padding and truncation to align dimensions.

### D.7 Additional Experimental Results

**Cross-domain transfer under domain shifts in transition dynamics.** To evaluate domain shifts with mismatches in transition dynamics, we conducted additional experiments using the Off-Dynamics Reinforcement Learning (ODRL) benchmark proposed by (Lyu et al., 2024b). This benchmark is explicitly designed to assess robustness under dynamics mismatches. In our experiments, we evaluated AdaptDICE in target domains with modified transition dynamics, including Hopper with gravity scaled to twice the original value and Ant with ground friction reduced to 0.1x the original. The results (Figure 8) show that AdaptDICE consistently outperforms baseline methods under these dynamics shifts.

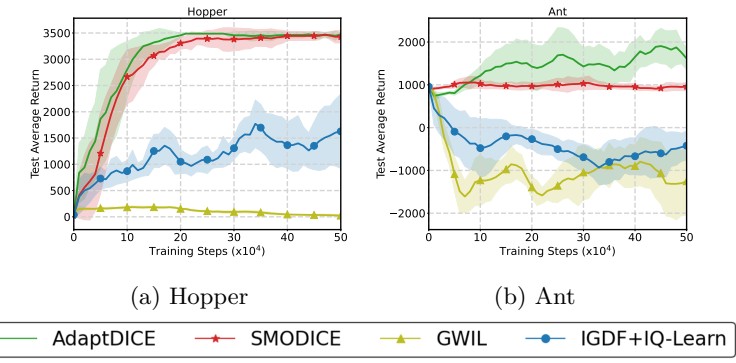

(a) Hopper          (b) Ant

Figure 8: Experiments on Hopper and Ant in the off-dynamics settings.

**Cross-domain transfer from only one source-domain expert trajectory.** In Figure 9, we evaluated AdaptDICE using only a single expert trajectory from the source domain. Notably, even in this low-data regime, AdaptDICE achieves competitive performance relative to GWIL, highlighting that its gains are not solely attributable to access to larger source datasets.

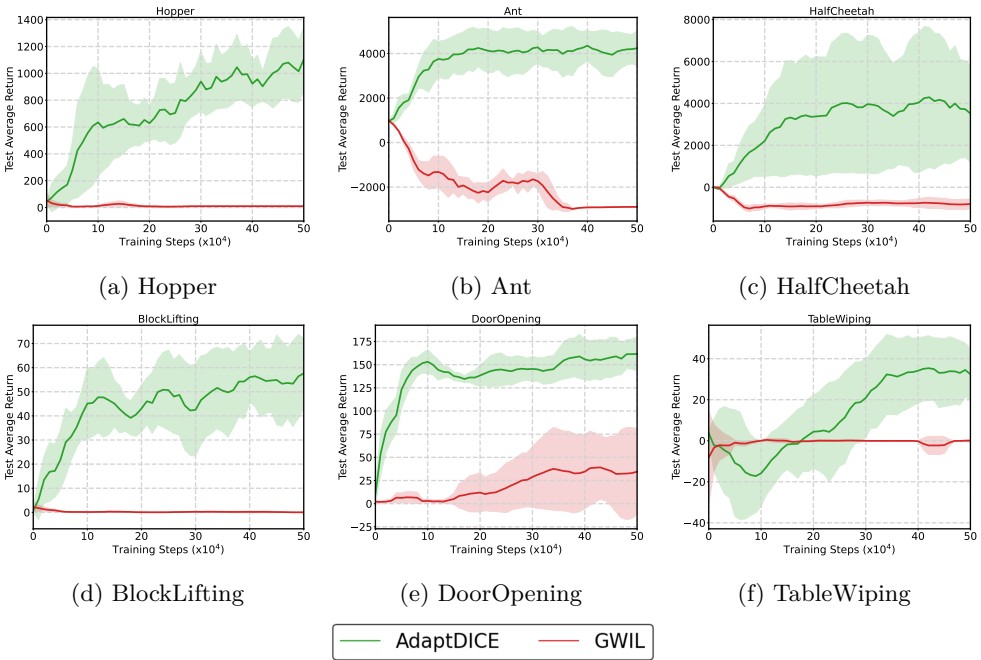

Figure 9: Performance comparison of AdaptDICE and GWIL with a single source-domain expert trajectory.

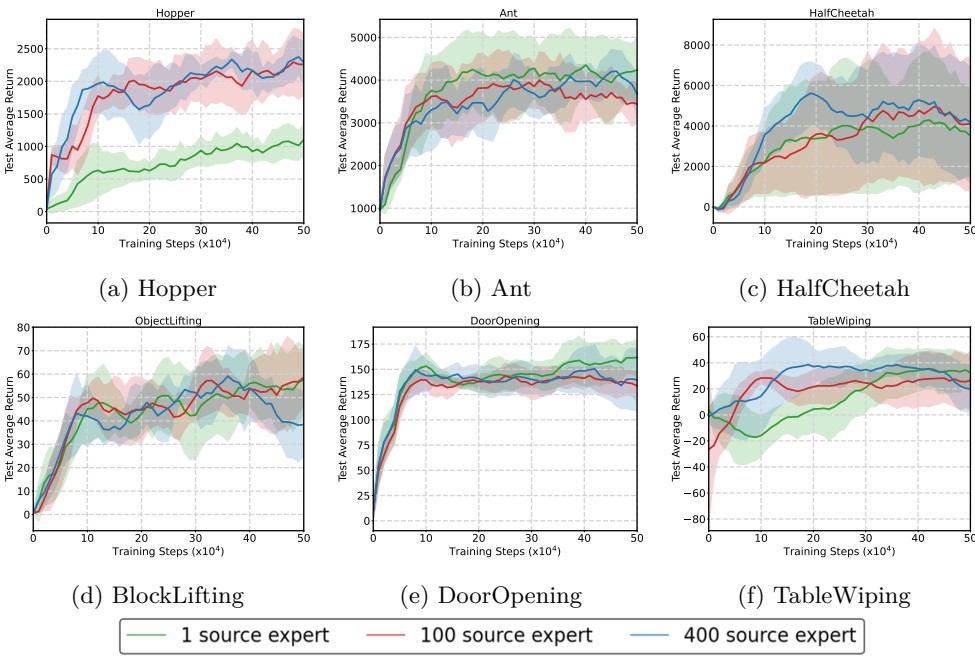

Figure 10: AdaptDICE's performance with pre-training with different amounts of source data (1, 100, 400 expert trajectories).

Table 11: The final performance of pre-trained models using different source-domain dataset configurations.

| Environment | 1 source expert trajectory | 100 source expert trajectories | 400 source expert trajectories |
|---|---|---|---|
| Hopper | 2384.1 | 3024.9 | **3568.6** |
| Ant | 1786.2 | 3262.2 | **5023.6** |
| HalfCheetah | $-5.7$ | 8233.0 | **11158.5** |
| BlockLifting | 28.4 | 165.7 | **193.5** |
| DoorOpening | 13.9 | 164.5 | **203.0** |
| TableWiping | 4.9 | 80.8 | **85.4** |

**Scaling effect of source-domain data.** In Figure 10, we conducted an empirical study on source data scaling across multiple environments, evaluating AdaptDICE with varying numbers of expert trajectories (1, 100, and 400). The results indicate that, in most environments, performance is relatively insensitive to the amount of source data. This behavior is consistent with the adaptive weighting mechanism in AdaptDICE, which down-weights unreliable source information and mitigates negative transfer.

**Adaptive $\beta$ versus fixed $\beta$.** We also compared the performance of AdaptDICE with our adaptive weighting scheme against that with fixed $\beta$ values (0.3, 0.5, and 0.7) in the Hopper and Ant environments. The results in Figure 11 show that smaller fixed $\beta$ values generally perform better under large domain discrepancies, as they bias learning toward the target-domain density ratio $w_{\text{tar}}$. However, the adaptive $\beta(t)$ consistently matches or outperforms the best fixed setting by dynamically adjusting throughout training. We also visualize the evolution of $\beta(t)$, confirming its adaptive behavior.

**Sensitivity to $\psi$.** We further evaluated the sensitivity to this parameter by testing $\psi \in \{0.5, 0.9, 0.99\}$ on Hopper and Ant ($\psi = 0.9$ is the default value used throughout the experiments). The results in Figure 12 indicate minimal performance variation across these values, supporting our claim that $\psi$ does not require task-specific tuning and does not function as a critical hyperparameter in practice.

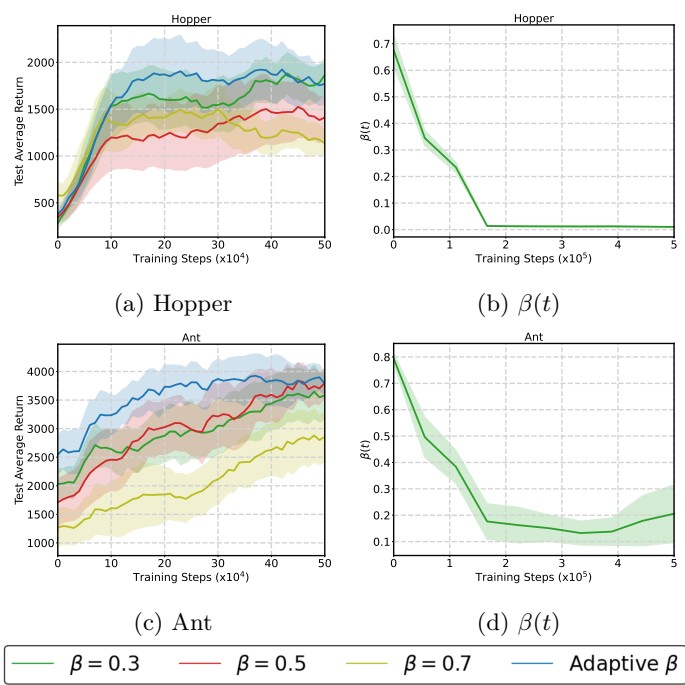

Figure 11: (a), (c): Comparison of AdaptDICE with adaptive $\beta$ and various fixed $\beta$ values (0.3, 0.5, and 0.7) on Hopper and Ant; (b), (d): Evolution of $\beta(t)$ under the adaptive weighting scheme on Hopper and Ant.

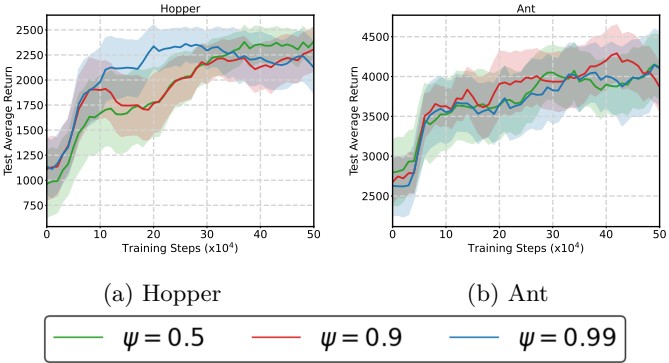

(a) Hopper  (b) Ant

Figure 12: Evaluation of AdatDICE with different $\psi$ values on Hopper and Ant.

**Loss curves of $L_{\mathbf{MAP}}$.** In Figure 13, we report the training curves of the mapping quality measured by the proposed mapping loss $L_{\mathrm{MAP}}$.

- In most environments, we observe a clear decrease in $L_{\mathrm{MAP}}$ over training, indicating progressively improved alignment between the source and target domains. These cases also coincide with the strong final performance of AdaptDICE, suggesting a positive correlation between improved mapping quality and successful transfer, which is consistent with the intuition behind Theorem 1.

- On the other hand, in Ant and HalfCheetah, the $L_{\mathrm{MAP}}$ values remain relatively flat and do not exhibit a clear downward trend. This behavior is closely aligned with the ablation study results reported in Appendix D.3 (Figure 3). In these environments, the source-domain component $w_{\mathrm{src}}$ fails to enable effective transfer and does not achieve meaningful performance when used alone (cf. the "Source-Only" curves in Figure 3). As a result, the source domain provides limited useful information to the target domain, which inherently limits both the improvement of the learned mappings and the potential benefits of cross-domain transfer.

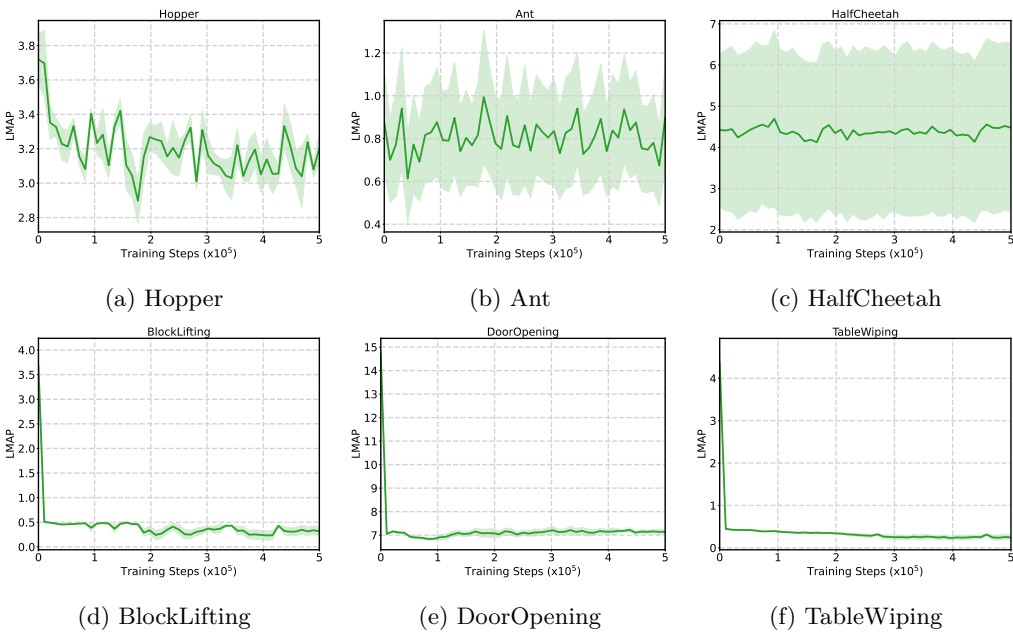

(a) Hopper  (b) Ant  (c) HalfCheetah

(d) BlockLifting  (e) DoorOpening  (f) TableWiping

Figure 13: Loss curves of $L_{\mathrm{MAP}}$.

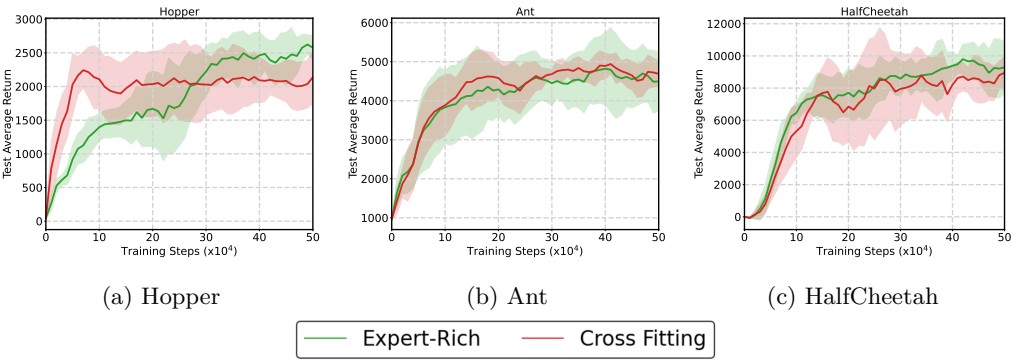

Figure 14: Experimental results of AdaptDICE with cross-fitting on $\beta(t)$ and $w_{\text{tar}}$.

**Calculation of $\hat{\beta}(t)$ through cross-fitting.** Recall from Equation (19) and Equation (21) that in practice $\hat{\beta}(t)$ needs to be estimated from state-action pairs and involves another stochastic estimate $w_{\text{tar}}$. To mitigate the potential circular dependencies, one enhancement is to decouple the selection of $\hat{\beta}(t)$ from the estimation of $w_{\text{tar}}^{(t)}$ through cross-fitting. In Figure 14, we conducted an additional experiment using cross-fitting: We split the target-domain dataset $\mathcal{D}_{\text{tar}}$ into two disjoint halves ($\mathcal{D}_1$ and $\mathcal{D}_2$). First, we estimated $w_{\text{src}}$ and $w_{\text{tar}}$ on $\mathcal{D}_1$, while computing the proxy errors $\Delta \hat{w}_{\text{src}}^{(t)}$ and $\Delta \hat{w}_{\text{tar}}^{(t)}$ (for $\beta(t)$) on the held-out $\mathcal{D}_2$. For the policy update via $w_{\text{cross}}$ (Equation (6)), we combined $\beta(t)$ from $\mathcal{D}_2$ with $w_{\text{src}}, w_{\text{tar}}$ from $\mathcal{D}_1$. Similarly, we also performed the reverse: estimating $w_{\text{src}}$ and $w_{\text{tar}}$ on $\mathcal{D}_2$ and $\beta(t)$ on $\mathcal{D}_1$. Then, we combine both losses to compute the final $L_{\text{BC}}$ (Equation (9)). This ensures $\beta(t)$ is validated on data that is not used in learning the density ratios.

We evaluate this variant of AdaptDICE on Hopper, Ant, and HalfCheetah under the Expert-Rich setting (which has more target-domain trajectories and hence is more appropriate for this data splitting than the Default setting). We observe that there appears to be improvement in the early training steps in some of the tasks (e.g., Hopper) and rather mild differences in the final performance between those with and without cross-fitting.

## D.8 Normalizing Flows for the Mapping Functions

Following (Brahmanage et al., 2023) the action-constrained RL literature (Lin et al., 2021; Hung et al., 2025), we employ normalizing flows to ensure that the outputs of $G$ and $H$ lie within the source-domain feasible region, without relying on projection that involves an online optimization procedure (e.g., quadratic program). Specifically:

- **Neural network architecture of $G$ and $H$**: In constructing the mapping functions $G$ and $H$, we use several fully-connected layers to first map the target-domain states/actions to some latent dummy region, followed by the normalizing flow model that transforms these latent vectors into the source-domain feasible region.

- **Latent dummy region**: We define a simple latent dummy region as a $N$-dimensional hypercube (e.g., $[0,1]^N$), where $N$ denotes the dimensionality. This latent region serves as the domain of the input distribution of the normalizing flow.

- **Flow model pre-training and usage**: We employ a conditional RealNVP flow consisting of six affine coupling layers with 256-unit MLPs. **The flow is pre-trained completely offline** by maximizing the log-likelihood of feasible samples, which are collected using Hamiltonian Monte Carlo (HMC). Pre-training is performed using Adam optimizer for 5,000-20,000 epochs, depending

on the environment. During target-domain learning, **the parameters of the flow model are kept fixed**, and only the weights of the prepended fully-connected layers are updated.

