# OpenReview forum: "Semi-Supervised Cross-Domain Imitation Learning"
_TMLR — Accepted by TMLR_

### Review · Reviewer_KGE7 · 2025-11-25

**Summary Of Contributions:**

1. The paper is generally well-written, with clear motivation and positioning, though the methodology section could benefit from improved exposition.

2. The experimental evaluation is extensive and thoughtfully designed, covering diverse domains and providing strong empirical support for the proposed method.

3. The proposed semi-supervised setting is both practical and impactful, addressing a key limitation in current cross-domain imitation learning approaches.

**Audience:**

Yes

**Audience Explanation:**

The semi-supervised Cross-Domain Imitation Learning (CDIL) setting addresses a practical and compelling problem with strong real-world relevance. It is likely to attract interest from the research community working on data-efficient imitation learning and transfer across domains.

**Claims And Evidence:**

Yes

**Claims Explanation:**

The paper’s claims are substantiated through rigorous theoretical analysis and comprehensive empirical validation.

**Requested Changes:**

1. It is recommended to use vector graphics (e.g., PDF or SVG) for figures such as Figure 1 to ensure clarity and scalability, especially for publication-quality rendering.

2. The authors state that the goal is to “learn a policy $\pi$ that can effectively mimic the expert policy based on data from both domains.” However, if the two domains differ significantly in their transition dynamics, the optimal policies in each domain may diverge substantially. How does the method reconcile these differences to produce a single effective policy?

3. In Algorithm 1, the definition of $\omega_{\text{src}}$ is missing. Additionally, the authors should reference the corresponding equations when introducing loss functions, for example, Line 2 should point to Equation (5), and Line 7 should clarify how $\beta(t)$ is computed. These adjustments would significantly enhance readability.

4. In Section 3.2, $H$ is defined as a mapping from $S_{\text{tar}} \times A_{\text{tar}}$ to $A_{\text{src}}$. The motivation and interpretation of this mapping are unclear. Why is it necessary to map target state-action pairs to source actions, rather than simply mapping $S_{\text{tar}}$ to $S_{\text{src}}$? A more intuitive explanation would be helpful.

5. The methodology section, particularly Section 3.2, is difficult to follow. The algorithm's components are introduced somewhat independently and abruptly. A clearer writing strategy that introduces each component sequentially, alongside their motivations and interdependencies, then an overall algorithm, would greatly improve comprehension.

6. In Section 4.1, the paper should explicitly describe the corresponding target-domain configurations (e.g., modified parameters or morphologies), either in the main text or in the appendix. This transparency is important for reproducibility and clarity.

7. The current experiments primarily evaluate domain shifts caused by changes in embodiment. It would strengthen the paper to also consider other types of domain shifts, such as changes to the transition function (e.g., added noise or even dynamics inversion), to test the generality of the proposed approach.

---

> ### Author Response · Authors · 2025-12-15
> **Author Responses to Reviewer KGE7 (1/3)**
>
> We thank the reviewer for the thoughtful and constructive feedback. We address each concern point-by-point below and have revised the paper accordingly.
>
> **[Requested Change 1]: Figure formats**
> > It is recommended to use vector graphics (e.g., PDF or SVG) for figures such as Figure 1 to ensure clarity and scalability, especially for publication-quality rendering.
>
> Following the reviewer’s recommendation, we have converted Figure 1 as well as all experimental plots to vector-based PDF formats to ensure clarity and publication-quality rendering. All updated figures are included in the revised manuscript. Thank you for the helpful suggestion.
>
> **[Requested Change 2]: Explain how AdaptDICE reconciles the domain differences**
> > The authors state that the goal is to “learn a policy $\pi$ that can effectively mimic the expert policy based on data from both domains.” However, if the two domains differ significantly in their transition dynamics, the optimal policies in each domain may diverge substantially. How does the method reconcile these differences to produce a single effective policy?
>
> Thank you for this thoughtful question, which highlights a fundamental challenge in cross-domain imitation learning.
>
> When the source and target domains differ substantially in their transition dynamics, their optimal policies may indeed diverge significantly. In such cases, the learned mapping functions $G$ and $H$ inevitably incur large alignment errors, leading to a large expected density-ratio error $\Delta\bar{w}^{(t)}\_{\text{src}}$​ for the mapped source estimator $w\_{\text{src}}(G(s),H(s, a))$ relative to the true target-domain optimal density ratio $w^{*}\_{\text{tar}}(s,a)$.
>
> Our adaptive weighting factor $\beta(t)$ (defined in Equation (18) and updated in Line 7 of Algorithm 1) is explicitly designed to detect and respond to this situation:$\beta(t) = \frac{\frac{1}{\Delta \bar{w}\_{\text{src}}^{(t)}}}{\frac{1}{\Delta \bar{w}\_{\text{src}}^{(t)}} + \frac{1}{\Delta \bar{w}\_{\text{tar}}^{(t)}}}.$
>
> When $\Delta \bar{w}^{(t)}\_{\text{src}} \gg \Delta \bar{w}^{(t)}\_{\text{tar}}$​—precisely the regime where cross-domain transfer becomes unreliable—$\beta(t)$ automatically approaches zero. As a result, the cross-domain density ratio reduces to
> $w\_{\text{cross}}(s,a) \approx w\_{\text{tar}}(s,a),$
> and the policy extraction objective in Eq. (9) simplifies to
> $L\_{\text{BC}}(\pi) \approx -\mathbb{E}_{(s,a)\sim D\_{\text{tar}}}[w\_{\text{tar}}(s,a)\log \pi(a|s)],$
> which is exactly the standard single-domain offline imitation learning objective on the target domain.
>
> Therefore, when domain discrepancies are too severe to support reliable transfer, our method gracefully degrades to pure target-domain learning, effectively discarding source-domain information and preventing negative transfer. This behavior is theoretically supported by Theorem 1 (Equation (17)), which bounds the expected cross-domain density-ratio error by $\min(\Delta \bar{w}^{(t)}\_{\text{src}}, \Delta \bar{w}^{(t)}\_{\text{tar}})$.
>
> In practice, this adaptive mechanism ensures that AdaptDICE never performs worse than a strong target-only baseline, while still exploiting source-domain knowledge whenever it is beneficial.
>
>
>
> **[Requested Change 3]: Presentation issues with Algorithm 1**
> > In Algorithm 1, the definition of $w_{\text{src}}$  is missing. Additionally, the authors should reference the corresponding equations when introducing loss functions, for example, Line 2 should point to Equation (5), and Line 7 should clarify how $\beta(t)$ is computed. These adjustments would significantly enhance readability.**
>
> Thank you for pointing out the presentation issues in Algorithm 1. We have revised the manuscript to improve clarity as follows.
>
> - The definition of $w_{\text{src}}$​ is explicitly provided in Equation (7), and we now clearly reference this equation when the term is used in the algorithm.
> - In addition, we have added explicit equation references for each loss function appearing in Algorithm 1. For example, Line 2 (discriminator update) now references the discriminator loss in Equation (10), and Line 7 explicitly refers to Equation (18) for the computation of the adaptive weight $\beta(t)$.
>
> These revisions are intended to improve readability and make the algorithm easier to follow.

---

> ### Author Response · Authors · 2025-12-15
> **Author Responses to Reviewer KGE7 (2/3)**
>
> **[Requested Change 4]: Explain why the mapping function $H$ is state-action-dependent**
> > In Section 3.2, $H$ is defined as a mapping from $S_{\text{tar}}\times A_{\text{tar}}$  to $A_{\text{src}}$. The motivation and interpretation of this mapping are unclear. Why is it necessary to map target state-action pairs to source actions, rather than simply mapping  $S_{\text{tar}}$ to $S_{\text{src}}$? A more intuitive explanation would be helpful.
>
> Thank you for the insightful question.
>
> While a pure state mapping $G: S_{\text{tar}} \rightarrow S_{\text{src}}$​ followed by policy composition is intuitively appealing, it implicitly assumes that actions have state-independent semantics. This assumption rarely holds in practice.
> The key reason we require a state-conditioned action mapping $H: S_{\text{tar}} \times A_{\text{tar}} \rightarrow A_{\text{src}}$ is that **the effect of an action can depend strongly on the current state**. For example, in robot control, due to kinematic and dynamic constraints, the same low-level action can induce very different outcomes under different situations, and conversely, achieving the same physical effect often requires different actions in different states.
>
> A state-independent action mapping would therefore be too restrictive to capture these situation-dependent semantics. By conditioning on $s_{\text{tar}}$​, $H$ can map $(s_{\text{tar}}, a_{\text{tar}})$ to the corresponding source-domain action that produces an equivalent physical effect under the aligned source-domain situation, enabling semantically meaningful action translation across domains with differing embodiments and dynamics.
>
> **[Requested Change 5]: Regarding the clarity of the methodology section**
> > The methodology section, particularly Section 3.2, is difficult to follow. The algorithm's components are introduced somewhat independently and abruptly. A clearer writing strategy that introduces each component sequentially, alongside their motivations and interdependencies, and then an overall algorithm, would greatly improve comprehension.
>
> We thank the reviewer for this insightful feedback regarding the clarity of the methodology presentation. We agree that a more sequential and structured exposition would improve readability.
> To address this concern, we have restructured Section 3.2 in the revised manuscript. We now begin with a high-level overview of the algorithm’s learnable components and their roles, explaining how the discriminator induces a pseudo reward that guides subsequent optimization.
>
> We then introduce the components in the order they are updated during training—namely, the cross-domain mapping loss, the DICE loss, and the cross-domain policy extraction objective—corresponding directly to Lines 8–10 of Algorithm 1. The pseudo reward computation, which is performed once before the training loop, is described afterward.
>
> Throughout the revised section, we explicitly emphasize the motivation for each component and clarify their interdependencies (e.g., how the pseudo reward influences both the mapping and DICE objectives, and how the learned mappings enable cross-domain density ratios for policy extraction). These changes reduce abrupt transitions and make the overall algorithmic flow more coherent.
>
> **[Requested Change 6]: Describe the target-domain configurations**
> > In Section 4.1, the paper should explicitly describe the corresponding target-domain configurations (e.g., modified parameters or morphologies), either in the main text or in the appendix. This transparency is important for reproducibility and clarity.
>
> We fully agree that explicitly describing the target-domain configurations is essential for reproducibility and clarity.
> To address this, we have added detailed documentation of all target-domain modifications in Appendix D.1 of the revised manuscript. This appendix includes:
> - Precise MuJoCo XML code snippets specifying the morphological changes for each task (e.g., adding an extra thigh to Hopper, a fifth leg to Ant, and a second back-leg set to HalfCheetah);
> - Descriptions of the Robosuite transfer settings (Panda → UR5e), with consistent object configurations and task goals, highlighting embodiment and kinematic mismatches;
> - A summary table of state and action dimensions for each source–target pair (Table 5);
> - Side-by-side visualizations of the source and target environments for intuitive comparison.
>
> Together, these additions ensure full transparency and enable precise replication of all experimental environments. We appreciate the reviewer’s emphasis on this point, which strengthens the overall rigor of the paper.

---

> ### Author Response · Authors · 2025-12-15
> **Author Responses to Reviewer KGE7 (3/3)**
>
> **[Requested Change 7]: Other types of domain shifts**
> > The current experiments primarily evaluate domain shifts caused by changes in embodiment. It would strengthen the paper also to consider other types of domain shifts, such as changes to the transition function (e.g., added noise or even dynamics inversion), to test the generality of the proposed approach.
>
> Thank you for this valuable suggestion. To evaluate domain shifts with mismatches in transition dynamics, we conducted additional experiments using the Off-Dynamics Reinforcement Learning (ODRL) benchmark proposed by Lyu et al. (2024).
> This benchmark is explicitly designed to assess robustness under dynamics mismatches. In our experiments, we evaluated AdaptDICE on target domains with modified transition dynamics, including Hopper with gravity scaled to twice the original value and Ant with ground friction reduced to 0.1x the original.
>
> The results show that AdaptDICE consistently outperforms baseline methods under these dynamics shifts. These results are included in the revised manuscript as Appendix D.7 (Figure 8), and are also available at: https://imgur.com/a/thcUWUH.

---

> > ### Comment · Reviewer_KGE7 · 2025-12-16
> >
> > I appreciate the authors’ thorough and clear responses. All of my concerns are satisfactorily resolved, and I consider this to be a solid and well-executed paper.

---

### Review · Reviewer_PiJH · 2025-11-30

**Summary Of Contributions:**

This paper introduces Semi-Supervised Cross-Domain Imitation Learning (SS-CDIL) to enable cross-domain policy transfer by combining small amounts of target-domain expert demonstrations with unlabeled “imperfect” trajectories. The authors propose AdaptDICE, an algorithmic framework for SS-CDIL, which learns cross-domain state-action mappings and adaptively combines source and target domain knowledge. The method is evaluated on MuJoCo locomotion and Robosuite manipulation tasks, demonstrating consistent improvements over baselines. While I have a few questions about the baseline comparisons and some parameter choices, the paper is clearly written, novel, and well-motivated.

**Additional Comments:**

I enjoyed reading the paper, it would be great to have access to the code to facilitate reproducibility and out of sample testing on data from different domains.

**Audience:**

Yes

**Audience Explanation:**

This paper is motivated by a very practical/prevalent constraint, which is that in many domains, expert demonstrations in the target environment are expensive to collect, while related unlabeled data is much easier to obtain. An approach that can make use of both will be appealing to a wide range of researchers across the TMLR audience, especially given the demonstrated improvements on a wide range of tasks. Overall, the combination of a new problem setup and a concrete, theoretically supported algorithm fits well within TMLR’s scope.

**Broader Impact Concerns:**

None. The work does not raise ethical concerns requiring a Broader Impact Statement.

**Claims And Evidence:**

Yes

**Claims Explanation:**

The main claims about AdaptDICE's performance are generally well-supported by performance on a wide range of tasks. The ablation studies effectively demonstrate that the approach outperforms using either source or target knowledge alone. However, some claims could be made stronger with additional support:

- The experimental comparison doesn’t fully disentangle algorithmic advantages from data availability advantages. AdaptDICE uses substantially more source data (400 expert trajectories, Table 7) than baselines (especially for GWIL, which uses 1 trajectory).

- The authors claim "hyperparameter-free adaptation", but then there is a moving average parameter psi=0.9, which looks like a hyperparameter. The paper also lacks ablation against intermediate fixed β values to demonstrate that adaptivity is necessary.

- The theoretical bound depends on the quality of the learned cross-domain mappings, but the paper does not empirically characterize what counts as low error, or how much performance degrades as error increases.

**Requested Changes:**

Critical for acceptance: None. The paper meets TMLR's acceptance criteria as-is.

There are a few additions that I think would strengthen the work:

1. The experimental setup raises fairness concerns. AdaptDICE uses 400 expert + 2000 sub-optimal source trajectories for pre-training, while GWIL uses only 1 source expert trajectory (a 400× difference), with other baselines falling in between. Several baselines are adapted from their original contexts, and some of the poor baseline performance may stem from these mismatches rather than fundamental algorithmic differences. The substantial source-data advantage should be more clearly acknowledged, and the authors could potentially add a source data scaling analysis. E.g., provide experiments showing AdaptDICE’s performance when pre-trained with different amounts of source data (e.g., 1, 10, 50, 100, 400 expert trajectories) to clarify whether the gains stem from algorithmic design or simply from access to more source data.

2. While Table 3 ablates β = 0 and β = 1, several key design choices are not tested. The method is described as using “hyperparameter-free adaptation,” but β(t) depends on a moving-average parameter ψ = 0.9, which effectively acts as a hyperparameter. The paper does not compare against intermediate fixed β values, and it does not explore alternatives to the mapping loss or analyze the learned mappings G and H. It would be helpful to add experiments comparing adaptive β(t) to fixed values such as 0.3, 0.5, and 0.7, and show the trajectory of beta(t) during training. For ψ, it could be helpful to evaluate a few different values of ψ  or clarify why ψ  = 0.9 is not considered a hyperparameter.

3. Theorem 1 provides a convergence guarantee, but its practical implications are unclear. The bound depends on the quality of the learned cross-domain mappings, yet the paper does not evaluate when these mappings are accurate or how performance changes as the mapping quality worsens. Without empirical characterization, it is hard for practitioners to assess when AdaptDICE is likely to help or when domain mismatch will limit its utility. It would be helpful if the authors could report the LMAP values achieved in each environment and examine whether they correlate with final performance.

---

> ### Author Response · Authors · 2025-12-15
> **Author Responses to Reviewer PiJH (1/2)**
>
> We thank the reviewer for the thoughtful and constructive feedback. We address each concern point-by-point below and have revised the paper accordingly.
>
> **[Requested Change 1]: Regarding the effect of the amount of source data**
> > The experimental setup raises fairness concerns. AdaptDICE uses 400 expert + 2000 sub-optimal source trajectories for pre-training, while GWIL uses only 1 source expert trajectory (a 400× difference), with other baselines falling in between. Several baselines are adapted from their original contexts, and some of the poor baseline performance may stem from these mismatches rather than fundamental algorithmic differences. The substantial source-data advantage should be more clearly acknowledged, and the authors could potentially add a source data scaling analysis. E.g., provide experiments showing AdaptDICE’s performance when pre-trained with different amounts of source data (e.g., 1, 10, 50, 100, 400 expert trajectories) to clarify whether the gains stem from algorithmic design or simply from access to more source data.**
>
> Thank you for the insightful feedback. We acknowledge that access to additional source data may influence performance comparisons, and we appreciate the suggestion to conduct a source data scaling analysis. To address this concern, we performed additional experiments and incorporated the results into the revised manuscript.
>
> - First, to enable a direct comparison under equivalent source-data constraints, we evaluated **AdaptDICE using only a single expert trajectory from the source domain**. Notably, even in this low-data regime, AdaptDICE achieves competitive performance relative to GWIL, highlighting that its gains are not solely attributable to access to larger source datasets. The corresponding training curves are shown in Figure 9 of Appendix D.7 (also available at https://imgur.com/a/weLVD7i).
>
> - Second, we conducted a source data scaling study across multiple environments, evaluating AdaptDICE with varying numbers of expert trajectories (1, 100, and 400). The results indicate that, in most environments, performance is relatively insensitive to the amount of source data. This behavior is consistent with the adaptive weighting mechanism in AdaptDICE, which down-weights unreliable source information and mitigates negative transfer. These results are provided in Figure 10 of Appendix D.7 (https://imgur.com/a/koRGyGs).
>
> **[Requested Change 2]: Empirical study on $\beta$**
> > While Table 3 ablates $\beta = 0$ and $\beta = 1$, several key design choices are not tested. The method is described as using “hyperparameter-free adaptation,” but $\beta(t)$ depends on a moving-average parameter $\psi = 0.9$, which effectively acts as a hyperparameter. The paper does not compare against intermediate fixed $\beta$ values, and it does not explore alternatives to the mapping loss or analyze the learned mappings $G$ and $H$. It would be helpful to add experiments comparing adaptive $\beta(t)$ to fixed values such as 0.3, 0.5, and 0.7, and show the trajectory of $\beta(t)$ during training. For $\psi$, it could be helpful to evaluate a few different values of $\psi$ or clarify why $\psi = 0.9$ is not considered a hyperparameter.**
>
> Thank you for highlighting these important design choices. We appreciate the suggestion to further ablate our method and have conducted additional experiments to address your concerns.
>
> **Fixed $\beta$ values and evolution of $\beta(t)$ with time**
>
> We compared our adaptive weighting scheme against fixed $\beta$ values (0.3, 0.5, and 0.7) on the Hopper and Ant environments. The results show that smaller fixed $\beta$ values generally perform better under large domain discrepancies, as they bias learning toward the target-domain density ratio $w_{\text{tar}}$​. However, the adaptive $\beta(t)$ consistently matches or outperforms the best fixed setting by dynamically adjusting throughout training. We also visualize the evolution of $\beta(t)$, confirming its adaptive behavior. These results are included in Appendix D.7 (Figure 11) and are available at https://imgur.com/a/tGMuD2h.
>
> **Moving-average parameter $\psi$**
>
> We further evaluated the sensitivity to this parameter by testing $\psi \in \{0.5, 0.9, 0.99\}$ on Hopper and Ant ($\psi = 0.9$ is the default value used throughout the experiments). The results indicate minimal performance variation across these values, supporting our claim that $\psi$ does not require task-specific tuning and does not function as a critical hyperparameter in practice. These results are provided in Appendix D.7 (Figure 12) and at https://imgur.com/a/NdXMhZl.

---

> ### Author Response · Authors · 2025-12-15
> **Author Responses to Reviewer PiJH (2/2)**
>
> **[Requested Change 3]: Report $L_{\text{MAP}}$ and examine the correlation with final performance**
> > Theorem 1 provides a convergence guarantee, but its practical implications are unclear. The bound depends on the quality of the learned cross-domain mappings, yet the paper does not evaluate when these mappings are accurate or how performance changes as the mapping quality worsens. Without empirical characterization, it is hard for practitioners to assess when AdaptDICE is likely to help or when domain mismatch will limit its utility. It would be helpful if the authors could report the $L_{\text{MAP}}$ values achieved in each environment and examine whether they correlate with final performance.
>
> Thank you for the insightful question regarding the practical implications of Theorem 1 and the role of cross-domain mapping quality. We agree that empirically characterizing the learned mappings is important for understanding when AdaptDICE is likely to be effective and when domain mismatch may limit its utility.
> To address this concern, we report the training curves of the mapping quality measured by the proposed mapping loss $L_{\text{MAP}}$. The corresponding results are available at https://imgur.com/a/1HQ5UXh, and we also included them in Appendix D.7 (Figure 13).
>
> - In most environments, we observe a clear decrease in $L_{\text{MAP}}$ over training, indicating progressively improved alignment between the source and target domains. These cases also coincide with the strong final performance of AdaptDICE, suggesting a positive correlation between improved mapping quality and successful transfer, which is consistent with the intuition behind Theorem 1.
>
> - On the other hand, in Ant and HalfCheetah, the $L_{\text{MAP}}$ values remain relatively flat and do not exhibit a clear downward trend. This behavior is closely aligned with the ablation study results reported in Appendix D.3 (Figure 3) in the updated manuscript. In these environments, the source-domain component $w_{\text{src}}$ fails to enable effective transfer and does not achieve meaningful performance when used alone (cf. the “Source-Only” curves in Figure 3). As a result, the source domain provides limited useful information to the target domain, which inherently limits both the improvement of the learned mappings and the potential benefits of cross-domain transfer.
>
> Overall, these results clarify that AdaptDICE is most effective when the learned mappings successfully reduce cross-domain discrepancies, while its benefits are naturally limited when the source-domain transferability itself is weak.
>
> **[Additional Comments]: Access to the code**
> > I enjoyed reading the paper, it would be great to have access to the code to facilitate reproducibility and out of sample testing on data from different domains.
>
> We sincerely thank the reviewer for the positive feedback. To support transparency and facilitate reproducibility, we will publicly release the full codebase on GitHub after the paper is accepted. The repository will include all implementation code, experiment scripts, and documentation required to reproduce the results reported in the paper.
>
> We will also include a permanent link to the repository in the camera-ready version to ensure long-term accessibility for the community. We appreciate the reviewer’s encouragement and look forward to sharing the code with the community.

---

### Review · Reviewer_89eE · 2025-12-01

**Summary Of Contributions:**

This paper presents AdaptDICE, an algorithm for policy learning in the specific setting of semi-supervised cross-domain imitation learning (SS-CDIL), as introduced in the paper. The authors emphasize why CDIL is important, the two domains often available: source domain and target domain, the main approaches often taken: CDIL with proxy tasks(a supervised setting) and unsupervised CDIL,  and why SS-CDIL is an acceptable approach that gets the best of both worlds. In SS-CDIL, AdaptDICE has three main parts: a mapping loss, density ratios from the source and task domains, combined into a hybrid density ratio, and a weighting factor $\Beta$(t) that allows adaptive blending of the two domains.

The paper presents a theoretical convergence guarantee analysis for the density ratio in AdaptDICE, practical implementation guides for AdaptDICE, and then experiments on MuJoCo and Robosuite simulators benchmarking AdaptDICE against GWIL, SMODICE, and IGDF+IQ-Learn. Each of these three algorithm benchmarks is adjusted to work in the SS-CDIL setting before being compared with AdaptDICE. AdaptDICE is shown to overall outperform the three benchmarks across six tasks: Hopper, HalfCheetah, Ant, BlockLifting, DoorOpening, and TableWiping.

Key strengths of the paper include the theoretical proofs and derivations (including a convergence guarantee) in both the main paper and the appendix, and the breadth of experimental design and ablations. A primary weakness of the paper is the omission of certain tasks in some experiments, without clarification of why those choices were made.

**Additional Comments:**

On a minor note, the paper contains a few grammatical or formatting errors that need correction and would benefit from general proofreading.

**Audience:**

Yes

**Audience Explanation:**

The findings of this paper would be of interest to researchers and audience members in the fields of Imitation Learning (IL), Reinforcement Learning (RL), and Robot Learning.

**Broader Impact Concerns:**

The paper has no Broader Impact statement section.


Beyond the known concerns about training costs for machine learning projects, no broader, unique impact concern comes to mind.

**Claims And Evidence:**

Yes

**Claims Explanation:**

The paper robustly and clearly makes its case for why SS-CDIL is needed, supported by motivation, theoretical proofs, evaluations, and a literature review, and systematically builds the various components of AdaptDICE.

**Requested Changes:**

1. In the ablations of AdaptDICE (Section 4.3) and the effect of dataset configurations (Section 4.4), why are the Hopper and TableWiping tasks omitted? The authors should either include them or provide sufficient justification for why they are omitted, given that the prior evaluations included them.

2. Similarly, in Appendix D4, while considering the effect of Dataset Configurations for the benchmarks, Ant and Table Wiping are omitted. First, the resultant task set is not the same as in Section 4.4. Secondly, authors should either include the missing tasks or provide sufficient justification for omitting them.

3. The authors should say more about how they made the design choices for the changes in the target domains (versus the source domain MuJoco. Zhang et al. (2021) only include the HalfCheetah task. How did the authors determine what to change across domains for the other two tasks, and do we know if these choices affect the algorithm performance?

---

> ### Author Response · Authors · 2025-12-15
> **Author Responses to Reviewer 89eE**
>
> We greatly appreciate the reviewer for the thoughtful and constructive feedback. We address each requested change point-by-point below and have revised the paper accordingly.
>
> **[Requested Change 1]: Regarding the ablation results of AdaptDICE**
> > In the ablations of AdaptDICE (Section 4.3) and the effect of dataset configurations (Section 4.4), why are the Hopper and TableWiping tasks omitted? The authors should either include them or provide sufficient justification for why they are omitted, given that the prior evaluations included them.
>
> Thank you for the suggestion. We have conducted the additional experiments and incorporated the results into the revised manuscript.
>
> Specifically, for the ablation study, we added the training curves for Hopper and TableWiping in Appendix D.3 (Figure 3) and updated Table 3 with the corresponding performance metrics. For the dataset configuration analysis, we included the corresponding training curves in Appendix D.3 (Figure 4) and updated Table 4 accordingly.
>
> These additional results confirm that the trends observed in the other tasks are consistent for Hopper and TableWiping as well, further strengthening our empirical findings.
>
> **[Requested Change 2]: Experiments about data configurations in appendix**
> > Similarly, in Appendix D4, while considering the effect of Dataset Configurations for the benchmarks, Ant and Table Wiping are omitted. First, the resultant task set is not the same as in Section 4.4. Secondly, authors should either include the missing tasks or provide sufficient justification for omitting them.
>
> Thank you for the suggestion. We have now completed the missing experiments and updated the revised manuscript accordingly. The results for Ant and TableWiping are included in Appendix D.4 (Figures 5–7), ensuring consistency between the task sets used in the main text and the appendix.
>
> **[Requested Change 3]: Explain the design choices for the changes in the target domains**
> > The authors should say more about how they made the design choices for the changes in the target domains (versus the source domain MuJoco. Zhang et al. (2021) only include the HalfCheetah task. How did the authors determine what to change across domains for the other two tasks, and do we know if these choices affect the algorithm performance?
>
> We thank the reviewer for the thoughtful suggestion. Providing additional context for the design choices of the target-domain modifications is indeed important for understanding the experimental setup.
>
> To address this, we expanded Appendix D.1 (“Settings of Evaluation Environments”) in the revised manuscript with a clearer explanation of how the target-domain changes were selected.
>
> - For HalfCheetah, we follow the morphological modification introduced by Zhang et al. (2021), adding a second back-leg set to create embodiment mismatches while preserving the original locomotion objective.
> - For Ant, we introduce a fifth “middle” leg, inspired by the cross-domain transfer settings in (Zhu et al., 2024), which similarly modifies agent morphology to induce representation and dynamics gaps.
> - For Hopper, we add an extra thigh, following analogous modifications used in (Xu et al., 2023).
>
> These modifications were chosen to systematically introduce comparable levels of cross-domain discrepancy, such as increased state/action dimensionality and altered dynamics, keeping task goals unchanged (summarized in Table 5). This design allows for fair evaluation of cross-domain transfer without confounding changes in objectives.
>
> Regarding their impact on performance, although we do not conduct ablations over different modification variants, the results in Table 2 show that AdaptDICE consistently outperforms baseline methods across all tasks, suggesting robustness to these standard morphological changes.
>
> References
> - Zhang et al., “Learning cross-domain correspondence for control with dynamics cycle-consistency,” ICLR 2021.
> - Zhu et al., "Cross domain policy transfer with effect cycle-consistency," ICRA  2024.
> - Xu et al., "Cross-domain policy adaptation via value-guided data filtering," NeurIPS 2023.
>
> **[Additional Comments]: Grammatical or formatting errors**
> > On a minor note, the paper contains a few grammatical or formatting errors that need correction and would benefit from general proofreading.
>
> Thank you for pointing this out. We have carefully proofread the revised manuscript and corrected grammatical and formatting issues throughout, and we expect the overall writing quality has been further improved accordingly.

---

> > ### Comment · Reviewer_89eE · 2025-12-27
> >
> > I thank the authors for their responses.
> >
> > My main concerns and requested changes have been satisfactorily addressed and I think the paper is in a much stronger position now.

---

### Review · Reviewer_QVu5 · 2025-12-13

**Summary Of Contributions:**

I think the main contribution of this work is a new machine learning setting, i.e., SS-CDIL, that formalizes semi-supervised cross-domain imitation learning where you have a small set of target expert demos plus many unlabeled/imperfect target trajectories, and optionally source-domain offline data.

Other contributions include:
- Algorithm (AdaptDICE): proposes an offline CDIL method that extends DICE-style distribution correction to cross-domain transfer with state/action/dynamics mismatch using (i) a cross-domain Bellman-consistency mapping loss + a hybrid density ratio for policy extraction + adaptive weighting.
- Practical recipe: implements mappings via normalizing flows to hit "feasible regions" and uses a moving-average estimator for the adaptive weighting in practice.

**Audience:**

Yes

**Audience Explanation:**

This work will be of interest to Imitation Learning and Robot Learning researchers.

**Broader Impact Concerns:**

No pressing concerns on ethics and societal impact come to my mind, but I recommend that the authors include a broader impact statement section.

**Claims And Evidence:**

Yes

**Claims Explanation:**

The main claims are mostly backed by the evidence provided. 2 minor points need scoping:
- "without requiring...assumptions on isomorphism": while you don’t assume explicit isomorphism, learning G and H implicitly assumes there exists some mapping that makes Bellman consistency meaningful and that your function classes/optimization can find it. That’s still an assumption, just weaker/implicit.
- "hyperparameter-free adaptation" is misleading: there are still many knobs (network architectures, optimizers, learning rates, flow training, smoothing, DICE temperature/regularization, etc)

**Requested Changes:**

- Replace "hyperparameter-free" with "no manual tuning of $\beta$" (or self-tuned mixing?) and explicitly list remaining hyperparameters.
- Decouple $\beta(t)$ selection from $w_{\mathrm{tar}}^{(t)}$ by computing the proxy error on a held-out target split (or via cross-fitting/ensemble disagreement), so $\beta(t)$ isn’t “validated” by the same estimate it helps produce.
- State explicitly that transfer is via querying source signals on mapped tuples $(G(s_{\mathrm{tar}}), H(s_{\mathrm{tar}},a_{\mathrm{tar}}))$ (i.e., using $w_{\mathrm{src}}$/$Q_{\mathrm{src}}$ as priors) while $w_{\mathrm{tar}}$ and $(G,H)$ still must be learned from target data to avoid degenerate mappings.

---

> ### Author Response · Authors · 2025-12-27
> **Author Responses to Reviewer QVu5**
>
> We thank the reviewer for the thoughtful and constructive feedback. We address each issue point-by-point below and have revised the paper accordingly.
>
> **[Requested Change 1]: Regarding the use of "hyperparameter-free"**
> > Replace "hyperparameter-free" with "no manual tuning of " (or self-tuned mixing?) and explicitly list remaining hyperparameters.
>
> Thank you for the helpful suggestion. We have addressed the reviewer’s concern regarding the use of the term “hyperparameter-free” by replacing all occurrences of this term throughout the manuscript. We now use the more precise wording, such as “no manual tuning of”, depending on the context. This clarification emphasizes that while our method still contains standard hyperparameters shared with prior work, it does *not* require manual tuning for the key components highlighted in the paper.
>
> To ensure full transparency, we explicitly list all remaining hyperparameters and computational details in Appendix D.5 (i.e., “Computational Resources and Hyperparameters”), including discount factors, regularization coefficients, network architectures, optimization settings, and training configurations (see Table 10).
>
> **[Requested Change 2]: Regarding the data used in calculating $\beta(t)$**
> > Decouple $\beta(t)$ selection from $w\_{\text{tar}}^{(t)}$ by computing the proxy error on a held-out target split (or via cross-fitting/ensemble disagreement), so $\beta(t)$ isn’t “validated” by the same estimate it helps produce.
>
> Thank you for the insightful suggestion. We agree that decoupling the selection of $\beta(t)$ from the estimation of $w\_{\text{tar}}^{(t)}$​ could mitigate the potential circular dependencies in the weighting mechanism.
>
> To address this, we conducted an additional experiment using cross-fitting:
> - We split the target-domain dataset $\mathcal{D}\_{\text{tar}}$ into two disjoint halves ($\mathcal{D}\_1$ and $\mathcal{D}\_2$).
> - First, we estimated $w\_{\text{src}}$ and $w\_{\text{tar}}$ on $\mathcal{D}\_1$, while computing the proxy errors $\Delta \hat{w}\_{\text{src}}^{(t)}$ and $\Delta \hat{w}\_{\text{tar}}^{(t)}$ (for $\beta(t)$) on the held-out $\mathcal{D}\_2$. For the policy update via $w\_{\text{cross}}$ (Equation (6)), we combined $\beta(t)$ from $\mathcal{D}\_2$ with $w\_{\text{src}}, w\_{\text{tar}}$ from $\mathcal{D}\_1$.
> - Similarly, we also performed the reverse: estimating $w\_{\text{src}}$ and $w\_{\text{tar}}$ on $\mathcal{D}\_2$ and $\beta(t)$ on $\mathcal{D}\_1$.
> - Then, we combine both losses to compute the final $L\_{\text{BC}}$ (Equation (9)). This ensures $\beta(t)$ is validated on data that is not used in learning the density ratios.
>
> We evaluated this on Hopper, Ant, and HalfCheetah under the Expert-Rich setting (which has more target-domain trajectories and hence is more appropriate for this data splitting than the Default setting). These results are available at https://imgur.com/a/f6FpSt3.
>
> We observe that there appears to be improvement in the early training steps in some of the tasks (e.g., Hopper) and rather mild differences in the final performance between those with and without cross-fitting. We have also included the results in Appendix D.7 (Figure 14) to describe the benefit of this cross-fitting scheme in selecting $\beta(t)$.
>
> **[Requested Change 3]: State explicitly how the transfer mechanism works**
> > State explicitly that transfer is via querying source signals on mapped tuples $(G(s\_{\text{tar}}), H(s\_{\text{tar}}, a\_{\text{tar}}))$ (i.e., using $w\_{\text{src}}/Q\_{\text{src}}$ as priors) while $w\_{\text{tar}}$ and $(G, H)$ still must be learned from target data to avoid degenerate mappings.
>
> Following the reviewer’s thoughtful suggestion, we have revised Section 3.2 (Proposed Algorithm) to clarify the mechanism of knowledge transfer.
>
> Specifically:
> - In the revised manuscript, we now state explicitly that transfer is performed by querying source-domain signals on mapped target tuples, i.e., $(G(s\_{\text{tar}}), H(s\_{\text{tar}}, a\_{\text{tar}}))$, where the source density ratio or value function (i.e., $w_{\text{src}} $ or $Q_{\text{src}}$​) serves as a prior.
> - At the same time, we clearly emphasize that both the mapping functions $G$, $H$ and the target density ratio $w_{\text{tar}}$​ are learned solely from the target-domain data, which is necessary to avoid learning degenerate mappings.
>
> Thank you for the helpful suggestions.

---

### Review · Reviewer_MikH · 2025-12-19

**Summary Of Contributions:**

The paper introduces a new algorithm named AdaptDICE (Adaptive Cross-Domain Imitation Learning with Distribution Correction Estimation) to address the current challenges of Cross-Domain Imitation Learning (CDIL), specifically on "Semi-Supervised" setting (SS-CDIL) where the target domain has very limited labeled expert demonstrations but access to unlabeled, imperfect trajectories.

The proposed AdaptDICE method extends the offline imitation learning method DemoDICE to the cross-domain setting using three main components:
- Cross-Domain Mapping Loss ($L_{MAP}$): A loss function that learns state ($G$) and action ($H$) mappings by minimizing the Bellman error in the mapped source-domain space, utilizing a pre-trained source critic ($Q_{src}$).
- Hybrid Density Ratio: The extraction of the target policy utilizes a hybrid density ratio ($w_{cross}$) that combines the pre-trained source density ratio and the learned target density ratio.
- Adaptive Weighting ($\beta(t)$): An adaptive mechanism that dynamically balances the influence of the source and target domains based on their relative estimation errors, removing the need for manual tuning.

**Additional Comments:**

Overall, the paper is well-structured and the complex methodological contributions are easy to follow.

**Audience:**

Yes

**Audience Explanation:**

The paper is relevant to researchers working on reinforcement learning, imitation learning, robotics and/or transfer learning.

**Broader Impact Concerns:**

None - the paper does not leverage any sensitive data or explicitly enable harmful dual-use applications beyond general control capabilities.

**Claims And Evidence:**

Yes

**Claims Explanation:**

The paper presents sufficient evidence through theoretical derivations and empirical evaluation to support its claims:
- The central claim that AdaptDICE works with minimal supervision is strongly supported by Table 2, where the method significantly outperforms baselines (SMODICE, GWIL) in various environments using the "Default" dataset setting (limited target expert data).
- The ablation study in Section 4.3 and Table 3 clearly isolates the contribution of the hybrid approach. The results show that using only the source density ratio ($w_{src}$) or only the target density ratio ($w_{tar}$) leads to performance collapse or suboptimal returns in complex tasks, validating Eq. (6)
- The claim that the adaptive weighting factor $\beta(t)$ stabilizes training and outperforms fixed weights is supported by Figure 11, which compares the adaptive method against fixed values, showing that the adaptive mechanism converges to the optimal weighting strategy dynamically.

**Requested Changes:**

- In Section 3.5, the authors mention using "normalizing flows" to ensure mapping functions $G$ and $H$ map to a feasible region15. This is a critical implementation detail that is currently under-explained. Please expand on how the dummy region is defined and how the flow is pre-trained or updated during the main loop.
- The method relies on a pre-trained source-domain pseudo Q-function $Q_{src}$16. While the paper assumes this is "optimal," in practice, source policies may be suboptimal. A brief discussion or a small experiment on how AdaptDICE behaves if $Q_{src}$ is noisy would add significant value.

---

> ### Author Response · Authors · 2025-12-27
> **Author Responses to Reviewer MikH**
>
> We thank the reviewer for these constructive comments and for highlighting several important aspects of our method. We address each issue point-by-point below and have revised the paper accordingly.
>
> **[Requested Change 1]: Elaborate on the use of normalizing flows in the mapping functions**
> > In Section 3.5, the authors mention using "normalizing flows" to ensure mapping functions $G$ and $H$ map to a feasible region. This is a critical implementation detail that is currently under-explained. Please expand on how the dummy region is defined and how the flow is pre-trained or updated during the main loop.
>
> Thank you for highlighting this important implementation design. Following the action-constrained RL literature (Brahmanag et al., 2023), we employ normalizing flows to ensure that the outputs of $G$ and $H$ lie within the source-domain feasible region, without relying on projection that involves an online optimization procedure (e.g., quadratic program). Specifically:
>
> **Neural network architecture of $G$ and $H$**: In constructing the mapping functions $G$ and $H$, we use several fully-connected layers to first map the target-domain states/actions to some latent dummy region, followed by the normalizing flow model that transforms these latent vectors into the source-domain feasible region.
>
> **Latent dummy region**: We define a simple latent dummy region as a $N$-dimensional hypercube (e.g., $[0,1]^N$), where $N$ denotes the dimensionality. This latent region serves as the domain of the input distribution of the normalizing flow.
>
> **Flow pre-training and usage**: We employ a conditional RealNVP flow consisting of six affine coupling layers with 256-unit MLPs. **The flow is pre-trained completely offline** by maximizing the log-likelihood of feasible samples, which are collected using Hamiltonian Monte Carlo (HMC). Pre-training is performed using Adam optimizer for 5,000-20,000 epochs, depending on the environment. During target-domain learning, **the parameters of the flow model are kept fixed**, and only the weights of the prepended fully-connected layers are updated. We have expanded Section 3.5 accordingly and included the above description in the updated manuscript.
>
> Reference
> -  Brahmanag et al., “FlowPG: Action-constrained policy gradient with normalizing flows,” NeurIPS 2023.
>
> **[Requested Change 2]: Regarding the quality of the source-domain pseudo Q-function**
> > The method relies on a pre-trained source-domain pseudo Q-function $Q\_{\text{src}}$. While the paper assumes this is "optimal," in practice, source policies may be suboptimal. A brief discussion or a small experiment on how AdaptDICE behaves if $Q\_{\text{src}}$ is noisy would add significant value.
>
> Thank you for the thoughtful suggestion. To evaluate the robustness of AdaptDICE to suboptimal or noisy $Q\_{\text{src}}$​, we conducted an additional experiment to evaluate our method under source-domain pseudo Q-functions of various quality (trained from 1, 100, and 400 source-domain expert trajectories).
>
> Table 1 below (also included as Table 11 in the updated manuscript) further reports the corresponding performance of the pre-trained source models used to obtain $Q\_{\text{src}}$​. The corresponding training curves are available at https://imgur.com/a/koRGyGs (and also in Figure 10 of Appendix D.7 in the updated manuscript).
>
> Despite the variability in source quality, AdaptDICE consistently achieves strong performance in most of the environments. This is because the source-domain signals derived from $Q\_{\text{src}}$​ are used as priors rather than hard constraints. When $Q\_{\text{src}}$​ is noisy or unreliable, the adaptive weighting mechanism naturally reduces reliance on the source prior and shifts emphasis toward the target-domain density ratio $w\_{\text{tar}}$​, thereby mitigating the impact of suboptimal source-domain information. This demonstrates the practical robustness of AdaptDICE as it does not require the source $Q\_{\text{src}}$​ to be optimal.
>
> #### Table 1: The final performance of source-domain pre-trained models under different source-domain dataset configurations.
>
> |   Environment | 1 source expert trajectory | 100 source expert trajectories| 400 source expert trajectories  |
> |----------------------|-----------------|-------------------|--------------------|
> | Hopper               | 2384.1        | 3024.9          | **3568.6**  |
> | Ant                  | 1786.2        | 3262.2          | **5023.6**  |
> | HalfCheetah          | -5.7          | 8233.0          | **11158.5** |
> | BlockLifting         | 28.4          | 165.7          | **193.5**   |
> | DoorOpening          | 13.9          | 164.5          | **203.0**   |
> | TableWiping          | 4.9           | 80.8            | **85.4**   |

---

### Decision · Action_Editor_mS95 · 2026-01-08

**Recommendation:** Accept as is

**Audience:**

Yes

**Audience Explanation:**

The paper is certainly useful for the empirical RL research community as it provides one effective way to do transfer learning. Hence, the paper will be useful for the community.

**Claims And Evidence:**

Yes

**Claims Explanation:**

This paper proposes a semi-supervised cross-domain imitation learning algorithm for MDP. In particular, the paper proposed an algorithm with theoretical guarantees that it can transfer from a limited source expert demonstration to the target domain. The empirical results showed the efficacy of the proposed approach.

The paper proposes (i) a cross-domain mapping loss that learns cross-domain mapping functions by mapping the Bellman equation in the cross-domain space, (ii) the proposed approach AdaptDICE, derives the target domain state-action occupancy density ratio by cross-domain mapping, and (iii) the density ratio is modulated using a hyper-parameter $\beta$ to smooth out the estimation error. In particular, finding the mapping $G$ and $H$ that transfers the source domain to the target domain is novel, and interesting.


All the contributions are backed up by solid evidence, as acknowledged by the reviewers. I have gone over the paper, and I agree with the reviewers. Hence, I am recommending acceptance.